# Pathobiont-induced suppressive immune imprints thwart T cell vaccine responses

Irshad Ahmed Hajam [1], Chih-Ming Tsai[1], Cesia Gonzalez[1], Juan Raphael Caldera [1,2], María Lázaro Díez [1,3], Xin Du [1], April Aralar [1], Brian Lin [1], William Duong[1] & George Y. Liu [1,4] ✉

Pathobionts have evolved many strategies to coexist with the host, but how immune evasion mechanisms contribute to the difficulty of developing vaccines against pathobionts is unclear. Meanwhile, *Staphylococcus aureus* (SA) has resisted human vaccine development to date. Here we show that prior SA exposure induces non-protective CD4[+] T cell imprints, leading to the blunting of protective IsdB vaccine responses. Mechanistically, these SA-experienced CD4[+] T cells express IL-10, which is further amplified by vaccination and impedes vaccine protection by binding with IL-10Rα on CD4[+] T cell and inhibit IL-17A production. IL-10 also mediates cross-suppression of IsdB and sdrE multi-antigen vaccine. By contrast, the inefficiency of SA IsdB, IsdA and MntC vaccines can be overcome by co-treatment with adjuvants that promote IL-17A and IFN-γ responses. We thus propose that IL-10 secreting, SA-experienced CD4[+] T cell imprints represent a staphylococcal immune escaping mechanism that needs to be taken into consideration for future vaccine development.

Pathobionts have maintained a long-standing relationship with the human host by negotiating coexistence through various immune evading strategies. Many pathobionts have to date proven to be elusive targets of vaccination. Notable among them are the ESK(C)APE (*Enterococcus faecium, Staphylococcus aureus* (SA)*, Klebsiella pneumoniae, Clostridium difficile, Acinetobacter baumannii, Pseudomonas aeruginosa* and *Enterobacter* spp.) pathogens for which no vaccines as of yet exist[1,2]. Likely, vaccinations against pathobionts need to overcome pathobiont's immune evasive strategies to be successful[3–6]. Hence, for hard to generate vaccines, understanding of vaccine biology in the context of pathobiont's immune evasion mechanisms may be required for the development of a successful vaccine.

SA is a pathobiont that colonizes and infects humans from early infancy, with up to fifty percent of infants having been exposed to SA by the age of 6 months[7–10]. SA is also a major opportunistic pathogen that has been the target of vaccine research effort since 1903, that has led to the development of a diverse list of seemingly effective vaccines[11–13]. Remarkably, for unclear reasons, none of these experimental vaccines have been successful in clinical trials[14,15]. Recently, we offered an explanation for the conundrum[16]. We posited that prior exposure to SA as routinely occurs in humans leads to immune imprints that somehow suppress subsequent vaccinations. Simulating the failed phase III SA vaccine trial that targeted staphylococcal iron-regulated surface determinant B (IsdB) antigen[15], we showed that an IsdB vaccine that was otherwise protective in naïve mice, was ineffective when administered to mice previously infected with SA. We showed that SA infection induces largely non-protective humoral immune imprints that are preferentially recalled with IsdB vaccination, which thus leads to vaccine inefficacy. In that study, we demonstrated that non-neutralizing Fab and non-opsonic Fc domains were responsible for the ineffective antibody response[16].

Although these findings shed new light on why SA vaccines failed, it remains unclear how the pathobiont is able to induce broad non-protective imprints in the first place. Hence, we set out to determine the pathobiont mechanism that underlies the development of non-protective imprints and subsequent vaccine suppression. In this study,

[1]Department of Pediatrics, University of California San Diego, San Diego, CA 92093, USA. [2]Quest Diagnostics, 33608 Ortega Hwy., San Juan Capistrano, CA 92675, USA. [3]AIDS Research Institute (IrsiCaixa). VIRus Immune Escape and VACcine Design (VIRIEVAC) Universitary Hospital German Trias i Pujol Crta Canyet s/n 08916, Badalona, Barcelona, Spain. [4]Division of Infectious Diseases, Rady Children's Hospital, San Diego, CA 92123, USA. ✉e-mail: gyliu@health.ucsd.edu

we show that Alum/IsdB vaccine recalls CD4[+] T cells that secrete abundant IL-10. IL-10 in turn suppresses host-protective IL-17 responses, consistent with our prior finding of SA evasion of IL-17 responses through IL-10[17]. Staphylococcal vaccine interference can be overcome by applying potent IL-17-inducing adjuvants with vaccine antigens in SA-experienced hosts. Hence, the pathogen evasion mechanisms of IL-17 suppression can unlock important strategies for overcoming staphylococcal vaccine failures.

## Results

### SA induces suppressive CD4[+] T cell imprints that drive the loss of IsdB vaccine mediated T cell protection

Our prior study established the critical role of humoral immune imprints in IsdB vaccine interference[16]. Here, we investigated the impact of immune imprints on protective T cell responses to IsdB vaccine adjuvanted with Alum (AIsdB). We used an established murine model[16] consisting of three intraperitoneal (i.p.) injections of SA LAC (USA300 strain) administered at a weekly interval. We then vaccinated the SA-exposed mice or naïve control mice weekly for three weeks, challenged the mice one week later, and then measured tissue SA burden 20 h later (Fig. 1a).

First, to query the protective role of T cells induced by AIsdB vaccination, we vaccinated B cell deficient (muMt) mice and also performed adoptive CD3[+] T cell transfer from WT mice vaccinated with IsdB, then challenged the recipient mice with SA (Fig. 1a-c and Supplementary Fig. 1a–c). Vaccination of naive muMt mice with AlumIsdB (AIsdB) was protective against SA challenge (Fig. 1b and Supplementary Fig. 1b), as was the adoptive transfer of total splenic CD3[+] T cells from naïve AIsdB vaccinated mice (Fig. 1c and Supplementary Fig. 1c). Protection was notable in the spleen, kidney and peritoneum, and was augmented when tissue CFU burden was measured after 48 h of infection compared to 20 h (Supplementary

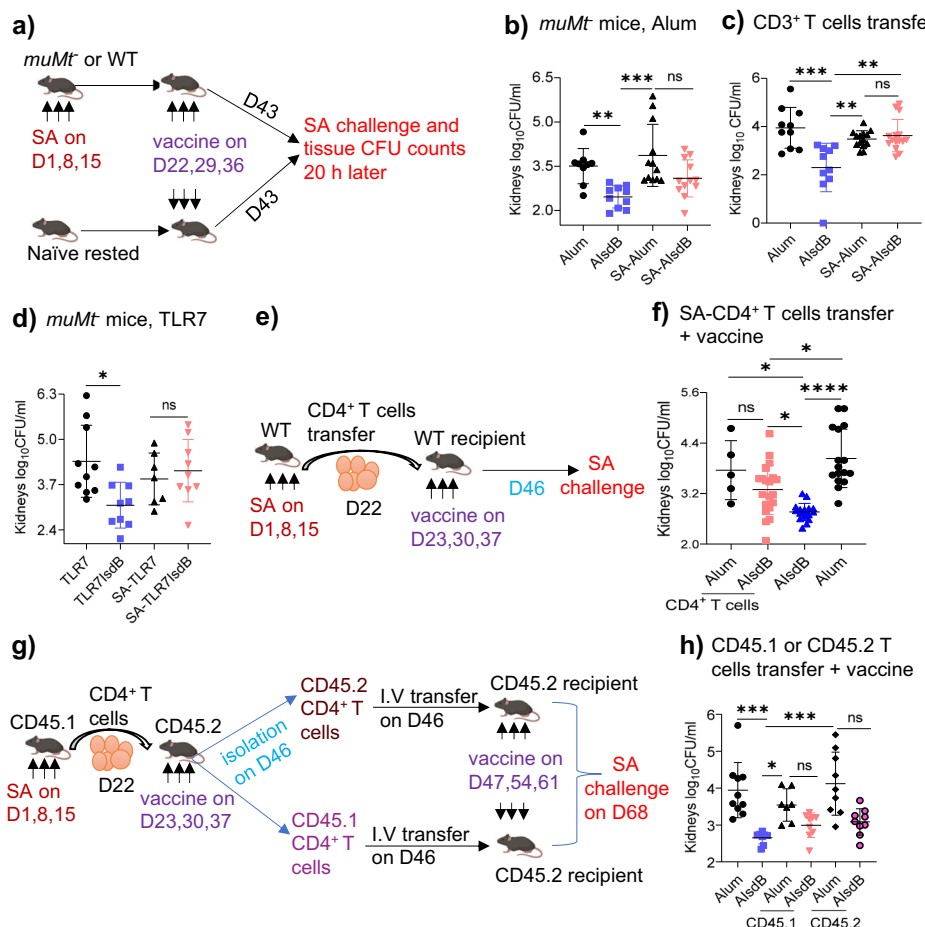

**Fig. 1 | SA T cell imprints abrogate T cell protection from IsdB vaccination.**
**a** Experimental setting. muMt or C57BL/6 wild-type (WT) mice were either rested or intraperitoneally (i.p.) infected with $3 \times 10^7$ SA (LAC) three times at 7-day (D) intervals. Then mice were vaccinated i.p. with either alum or AlumIsdB (AIsdB) and challenged i.p. with SA ($3 \times 10^7$) 7 days post-last vaccination (dpv). Bacterial burden in tissues was measured 20 h post-SA challenge. **b** Naïve or SA-exposed muMt mice were vaccinated with either Alum (n = 9 in naïve and 12 in SA-exposed) or AIsdB (n = 10 in naïve and 12 in SA-exposed), then SA challenged as in (**a**). **c** Naïve or SA-exposed WT mice were vaccinated with either Alum (n = 10 in naïve and 14 in SA-exposed) or AIsdB (n = 10 in naïve and 15 in SA-exposed) as in **a**. Splenic CD3[+] T cells isolated on 14 dpv were adoptively transferred into naïve recipient mice, followed by SA challenge. **d** Naïve or SA-exposed muMt mice were vaccinated with either TLR7 (Gardiquimod) alone (n = 10 in naïve and 7 in SA-exposed) or TLR7IsdB (n = 9 in naïve or SA-exposed), then SA challenged as in (**a**). **e**, **f** Splenic CD4[+] T cells ($1 \times 10^7$) isolated from SA-exposed WT mice were adoptively transferred into naïve recipient WT mice. One day later, the recipient mice were vaccinated with either Alum (n = 5 for Alum-CD4[+] T cells and 16 for Alum) or AIsdB (n = 20 for CD4[+] T AIsdB and 17 for AIsdB), followed by SA challenge as in (**a**). **g**, **h** SA-exposed splenic CD45.1 CD4[+] T cells ($1 \times 10^7$) were transferred into naïve CD45.2 mice followed by AIsdB vaccination. Splenic CD45.1 or CD45.2 CD4[+] T cells were isolated from the vaccinated mice and transferred into naive recipient mice, vaccinated with Alum (n = 10, 7, 9) or AIsdB (n = 8, 9, 9) and SA challenged as in (**a**). Data were from two to four independent experiments with each data point representing one mouse. The data are presented as mean ± SD of biological replicates, and were analyzed by Kruskal-Wallis non-parametric one-way ANOVA test. *p < 0.05, **p < 0.01, ***p < 0.001, ****p < 0.0001. ns, non-significant. dpv, days post-last vaccination. SA, Staphylococcus aureus. WT-wild-type. Source data are provided as Source Data File. Mouse image was created by BioRender (Created in BioRender. Hajam, I. (2024) https://BioRender.com/a18v205).

Fig. 1d–f), likely reflecting increased adaptive immune protection over time.

We investigated the CD3[+] T cell subset that conferred AIsdB vaccine protection by depleting T cell subsets using monoclonal antibodies (MAb). Anti-CD4 MAb treatment of WT mice prior to SA challenge abolished the protective efficacy of the AIsdB vaccine in the spleen, but not in the kidneys (Supplementary Fig. 1g). In comparison, treatment with either anti-CD8 Mab or anti-γδ TCR had no clear effect on either spleen or kidney protection. Thus, vaccine induced CD4[+] T cells appear to have a protective role in the spleen, but the specific subset of CD3[+] T cells (including the possibility of innate immune cells) that mediates protection in the kidneys is not clear.

To assess the effect of prior SA infection on vaccine-mediated protective T cell responses, we performed vaccination experiments as before in mice previously exposed to SA (Fig. 1a). In stark contrast to vaccination of naïve mice, both IsdB vaccination of SA-exposed *muMt*[-] mice (Fig. 1b and Supplementary Fig. 1b) and the adoptive transfer of splenic CD3[+] T cells, from SA-infected then vaccinated donors to naïve recipient mice, were non-protective against SA challenge (Fig. 1c and Supplementary Fig. 1c).

We next sought to determine if SA modulation of IsdB T cell vaccine responses occurs uniquely with Alum adjuvanted vaccines. We repeated IsdB vaccination of naïve and SA-exposed mice using a TLR7 agonist (Gardiquimod) adjuvant instead of alum. Consistent with prior results, TLR7 IsdB was effective in naïve *muMt*[-] mice but ineffective in *muMt*[-] mice previously infected with SA (Fig. 1d and Supplementary Fig. 1h). We thus conclude that prior SA exposure could effectively blunt protective T cell responses to IsdB vaccines.

To determine if T cell imprints from prior SA infections suppressed IsdB vaccine protection, we isolated splenic CD4[+] T cells from SA-infected mice (SA-CD4[+] T cells), then adoptively transferred the cells into naïve recipient WT mice, then vaccinated the mice with either Alum or AIsdB (Fig. 1e). Compared to naïve mice vaccinated with IsdB, mice given SA-CD4[+] T cells showed blunted protective response to AIsdB vaccine (Fig. 1f and Supplementary Fig. 1i), indicating that SA-CD4[+] T cell imprints play an active role in IsdB vaccine suppression.

Next, to determine if SA-CD4[+] T cells modify de novo T cell responses to IsdB vaccination, we performed adoptive transfer of SA-CD45.1 CD4[+] T cells into naïve CD45.2 mice, then vaccinated the recipients with AIsdB. 7 days (D) after the last vaccination, we isolated CD45.1 and CD45.2 CD4[+] T cells and injected the cells separately into naïve mice, then vaccinated the recipients with either Alum or AIsdB vaccination and challenged the mice with SA (Fig. 1g). Recipients of either CD45.1 or CD45.2 CD4[+] T cells had reduced protection from AIsdB vaccination, most notably in the kidneys (Fig. 1h and Supplementary Fig. 1j), suggesting that SA-CD4[+] T cells could potentially modify de novo T cell responses to IsdB vaccination. However,

the effect appears modest or not entirely clear cut in spite of significance based on statistics.

Taken together, our findings indicate that prior SA exposure negatively impact IsdB T cell vaccine responses. Particularly, SA exposure induces CD4[+] T cells that appear to drive the non-protective responses to IsdB vaccination.

## SA suppression of vaccine is CD4[+]CD25[+]IL-10[+] T cell mediated and IL-10 dependent

Various staphylococcal T modulatory mechanisms have been described, including toxin-mediated killing of CD4[+] T and antigen-presenting cells (APCs)[18–20], and the induction of immunomodulatory cytokine IL-10 by staphylococcal toxic shock syndrome toxin (TSST), phenol-soluble modulins (PSM), and O-acetyl transferase A (OatA)[17,21,22]. Abrogation of IsdB vaccine protection by transfer of CD4[+] T cells

(Fig. 1f) alone into naïve hosts unexposed to SA toxins suggests that vaccine interference occurs independently of T or APC depletion by toxins in our model. Hence, we next queried the role of IL-10 in IsdB vaccine suppression.

We measured ex vivo IL-10 recall responses to IsdB or heat-killed SA (HKB) in splenocytes isolated from previously unexposed or SA-exposed vaccinated mice. IL-10 levels increased with repeat SA infections (Supplementary Fig. 2a), and was significantly amplified with AIsdB vaccination compared with IL-10 levels from naïve mice vaccinated with AIsdB (Fig. 2a). Blockade of IL-10 in SA-exposed mice at the time of vaccination restored protective efficacy of the AIsdB vaccine (Fig. 2b, c, and Supplementary Fig. 2b), indicating that IL-10 is critical for the ablation of IsdB vaccine efficacy.

To investigate the adaptive immune sources of IL-10, we utilized IL-10[eGFP] (Vert-X) reporter mice and analyzed for *eGFP* (IL-10) in immune cells. Notably, only IL-10 expression in T and B cells is measured since autofluorescence is induced by the innate cells[23]. For the study, SA-exposed or non-exposed mice (NT) were challenged with SA and then analyzed for IL-10 expression 20 h later (Fig. 2d and Supplementary Fig. 2c). Although the frequencies of CD4[+] T, CD8[+] T and B cells did not change 20 h after SA infections (Supplementary Fig. 3a), repeat SA infections induced significantly higher expressions of IL-10 as measured by the *eGFP*[+] cells (Fig. 2e). Among the cell lineages, CD4[+] T cells expressed the highest level of IL-10 compared to CD8[+] and B cells based on flow-cytometry analysis and ELISPOT assay (Fig. 2e, f, and Supplementary Fig. 3b). With vaccination, IL-10 expression remained most dominant in CD4[+] T cells compared to B cells (Fig. 2g and Supplementary Fig. 3c). These findings suggest that IsdB vaccination induces robust IL-10 expression by adaptive immune cells, particularly CD4[+] T cells.

To further determine the vaccine suppressive role of IL-10 secreted selectively by SA-CD4[+] T cells, we performed adoptive transfer of CD4[+] T cells from SA-infected WT mice into naïve mice, then treated the recipient mice with either an isotype IgG1 or anti-IL-10 (aIL-10) followed by vaccination. The aIL-10 treatment abrogated vaccine suppression by SA-CD4[+] T cells (Supplementary Fig. 3d), thus strongly implicating IL-10 secreted by CD4[+] T cell imprints in the impairment of vaccine protective responses.

CD4[+]CD25[+][24,25] and CD4[+]CD69[+] T cells[26,27] are two regulatory cell lineages that exert IL-10 dependent regulatory functions. We measured IL-10 expression by these cell lineages in Vert-X mice infected with SA using flow cytometry (Fig. 2d). Although the percentage of CD4[+]CD69[+] and CD4[+]CD25[+] T cells increased with repeat SA infections, only CD4[+]CD25[+] T cells showed increased *eGFP* (IL-10) expressions with reinfections (Fig. 2h, i). Depletion of CD4[+]CD25[+] T cells by MAb 1D before AIsdB vaccination restored IsdB vaccine protection in SA-exposed WT mice (Fig. 2j and Supplementary Fig. 3e). Depletion of CD4[+]CD69[+] T cells trended towards restoration of vaccine protection, although significance was not reached. The CD69 receptor could be expressed on protective T effector cells and could be affected by the depleting antibodies. Hence, we conclude that at least CD4[+]CD25[+] T cells are critical in driving vaccine failure.

To address antigen-specificity of vaccine suppression, we infected mice with either wild-type (wt) or an isogenic IsdB/HarA ko strain of SA that lacks IsdB or IsdB homologue HarA[16]. We then adoptively transferred splenic SA-CD4[+] T cells into naïve mice, then vaccinated the recipients with Alum or AIsdB (Fig. 2k). Vaccination was non-protective in mice that received wt SA-CD4[+] T cells, and was protective when transferred CD4[+] T cells derived from mice infected with IsdB/HarA ko SA (Fig. 2l), with spleen data being more robust than kidney data. Altogether, our findings indicate that imprint mediated suppression of IsdB vaccine is at least CD4[+]CD25[+] T cell mediated, IL-10-dependent, and is suggestive to be antigen-specific.

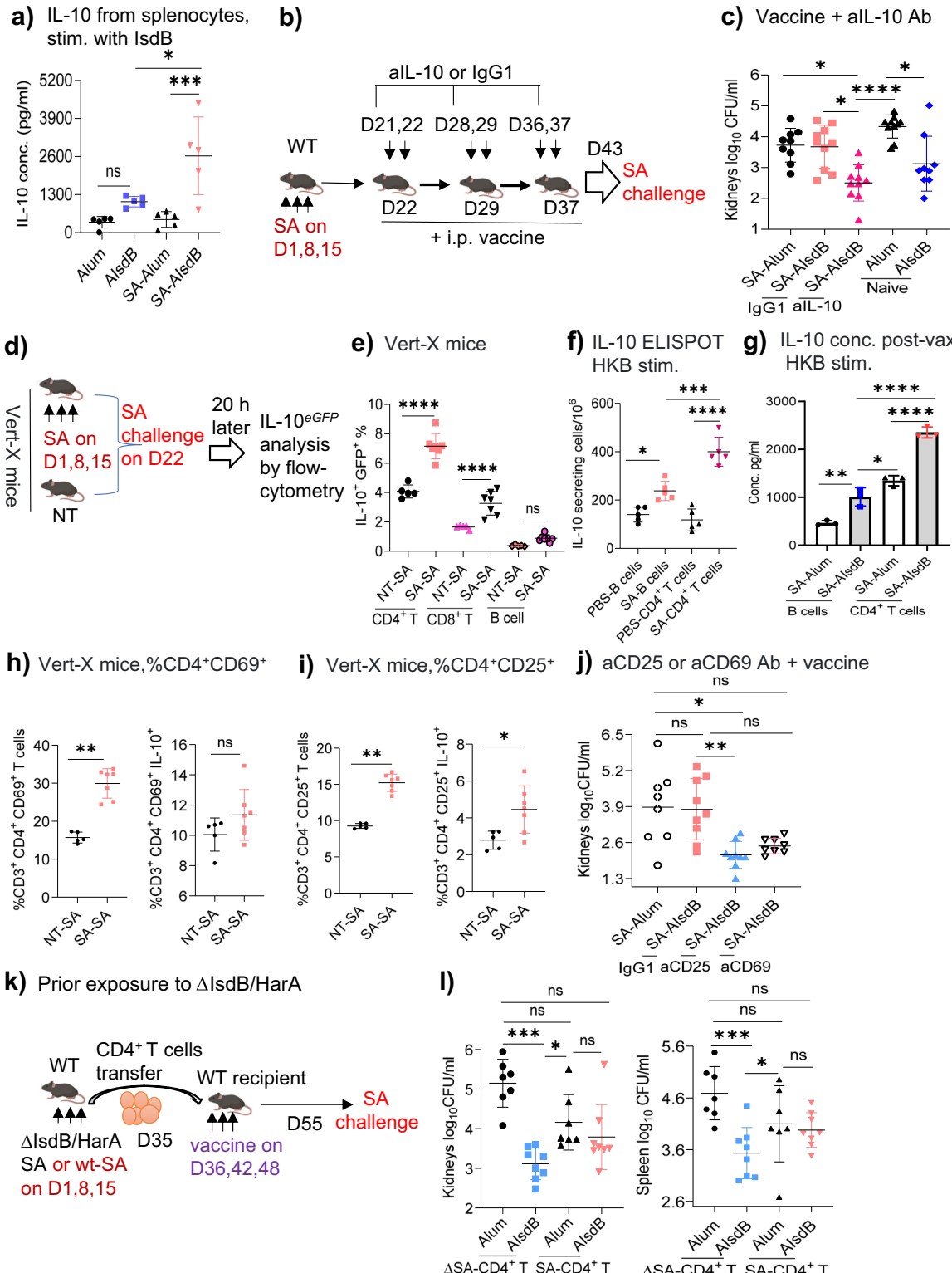

## IL-10 driven vaccine suppression is dependent on IL-10 and IL-10Rα expression by CD4+ T cells and STAT3 signaling

Although IL-10 is expressed most abundantly by CD4+ cells among adaptive immune cells tested, macrophages are a significant source of IL-10 in response to acute SA infection[28]. To determine the cellular source of IL-10 critical for vaccine interference, we interrogated AIsdB vaccine efficacy in mice with macrophage- (IL10^flox/flox x LysM^cre, referred to as MIL-10^-/-) and CD4+ T cell-specific deletion of IL-10 (IL10^flox/flox x CD4^Cre, referred to as CD4IL10^-/-). Consistent with the importance of IL-

10 derived from CD4+ T cells, AIsdB vaccination of CD4IL10^-/- mice pre-exposed to SA was protective. In comparison, IsdB vaccination remained ineffective in MIL-10^-/- mice (Fig. 3a, b and Supplementary Fig. 4a, b), suggesting a more important role of IL-10 secreting CD4+ T cells in driving vaccine suppression.

To investigate if CD4+ T cells are the target of IL-10 suppression, we vaccinated mice with specific deletion of IL-10Rα in CD4+ T cells (IL10Rα^flox/flox x CD4^Cre, referred to as CD4IL10Rα^-/- or +/+) after SA exposure. Although AIsdB vaccination of SA-exposed WT mice was not

**Fig. 2 | CD4⁺CD25⁺ IL-10⁺ T cells mediate IL10-dependent suppression of IsdB vaccine. a** Splenocytes from naïve- or SA-exposed vaccinated mice were isolated on 14dpv and stimulated with IsdB (10 μg/ml) for 60 h, followed by IL-10 measurement in the supernatant ($n = 5$ per group). **b, c** Mice were i.p. infected with SA ($3 \times 10^7$ CFU), then treated intravenously one day before and on the day of vaccination with either isotype IgG1 Ab ($n = 10$) or anti-IL-10 (aIL-10) MAb ($n = 10$). 7 dpv, mice were challenged with SA. Bacterial burden was measured 20 h post-challenge. For SA-Alum, Alum and AIsdB, n are 9, 10 and 9, respectively. **d** Experimental setting. Vert-X mice were untreated (NT) or exposed to SA, then SA challenged. Splenocytes were analyzed for eGFP (IL-10) expressions by flow cytometry 20 h later. **e** Analysis of eGFP (IL-10) expression by adaptive immune cells from naïve ($n = 5$) and SA-exposed ($n = 7$) Vert-X mice post-SA challenge. **f** Purified B or CD4⁺ T cells from PBS or SA-exposed WT mice were stimulated with naïve splenocytes plus HKB (1:10) for 40 h. IL-10 ELISPOT assay ($n = 5$ per group). **g** Pooled purified B ($n = 5$) or CD4⁺ T cells ($n = 5$) from SA-exposed vaccinated mice were stimulated with HKB (1:10) for 60 h. Culture supernatants were analyzed for IL-10. **h, i** Untreated ($n = 5$) or SA-exposed Verti-X mice ($n = 7$) were challenged with SA as shown in Fig. 2d. Analysis of splenic eGFP⁺ (IL-10) CD3⁺CD4⁺CD69⁺ T cells (**h**) and CD3⁺CD4⁺CD25⁺ T cells (**i**) 20 h

after SA infection. **j** SA-exposed WT mice were depleted of either CD25⁺ ($n = 9$) or CD69⁺ T cells ($n = 8$) one day before vaccination, then challenged with SA ($n = 8-9$ per group). For SA-Alum and SA-AIsdB, n are 8 and 9, respectively. **k, l** CD4⁺ T cells isolated from WT mice, previously exposed to either Baker WT (Alum, $n = 7$; AIsdB, $n = 8$) or IsdB/HarA double mutant (Alum, $n = 7$; AIsdB, $n = 8$), were adoptively transferred into naïve WT mice. The recipient mice were vaccinated, challenged with SA Baker as in Fig.1a. Data were from one to two independent experiments with each data point representing one mouse, except in (**g**). The data are presented as mean ± SD of biological replicates, except for (**g**) where the data are presented as mean ± SD of three technical replicates. Data in (**a, e–g**) were analyzed by one-way ANOVA with Tukey's posthoc test, data in (**h, i**) by two-tailed non-parametric Mann-Whitney T test, while the data in (**c, j–l**) were analyzed by Kruskal-Wallis non-parametric one-way ANOVA test. *$p < 0.05$, **$p < 0.01$, ***$p < 0.001$, ****$p < 0.0001$. ns, non-significant. dpv, days post-last vaccination. HKB, heat-killed bacteria. SA, *Staphylococcus aureus*. WT-wild-type. Source data are provided as Source Data File. Mouse image was created by BioRender (Created in BioRender. Hajam, I. (2024) https://BioRender.com/a18v205).

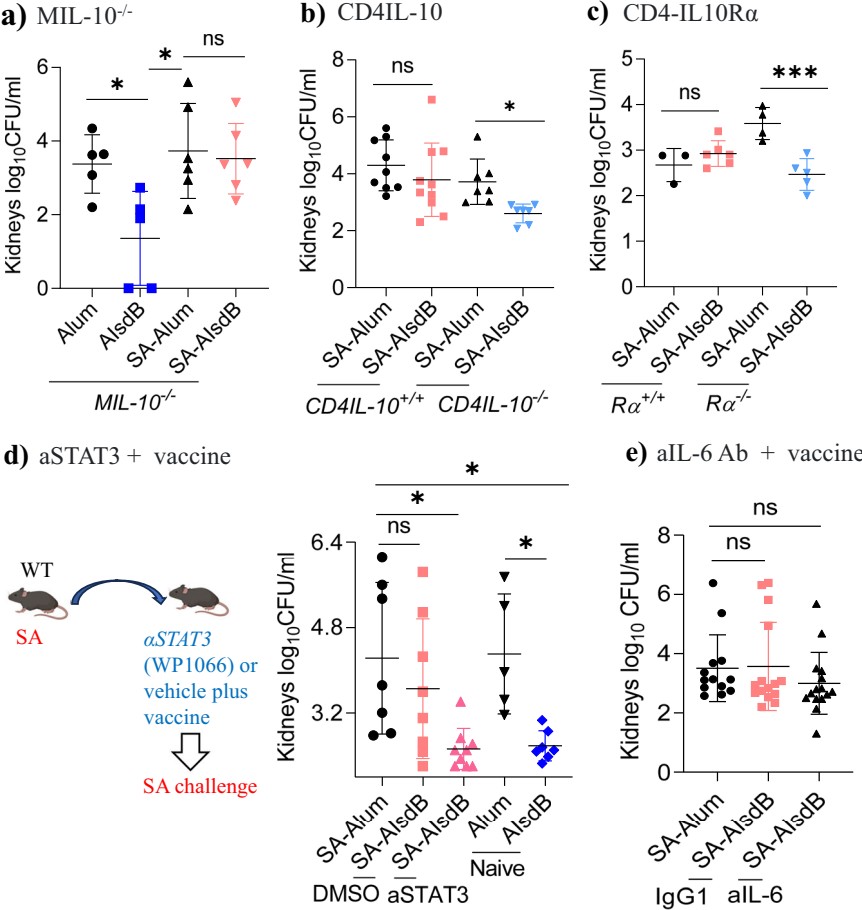

**Fig. 3 | IsdB vaccine suppression is dependent on STAT3 signaling, IL-10 and 10-Rα expression on CD4⁺ T cells. a** Naïve ($n = 5$) or SA-exposed ($n = 6$) MIL-10⁻/⁻ (IL10^flox/flox x LysM^cre) were vaccinated i.p. with either Alum or AIsdB, then challenged with SA on 7 dpv. **b** SA-exposed CD4IL-10⁺/⁺ or CD4IL-10⁻/⁻ were vaccinated i.p. with either Alum ($n = 9$ in CD4IL-10⁺/⁺ and 7 in CD4IL-10⁻/⁻) or AIsdB ($n = 10$ in CD4IL-10⁺/⁺ and 7 in CD4IL-10⁻/⁻), then challenged with SA on 7 dpv. **c** SA-exposed CD4IL10Rα⁺/⁺ or CD4IL10Rα⁻/⁻ were vaccinated i.p. with either Alum ($n = 3$ in Rα⁺/⁺ and 4 in Rα⁻/⁻) or AIsdB ($n = 6$ in Rα⁺/⁺ and 5 in Rα⁻/⁻), then challenged with SA on 7 dpv. **d** SA-exposed WT mice were treated 1 day before, on the day of vaccination and 1 day after with either vehicle (DMSO, $n = 8$) or an anti-STAT3 (aSTAT3) inhibitor ($n = 8$), then challenged with SA on 7 dpv. For SA-Alum, naïve Alum and naïve AIsdBsdrE, n are 7,

6 and 7, respectively. **e** SA-exposed WT mice were treated one day before and on the day of vaccination with either isotype IgG1 control ($n = 15$) or anti-IL-6 (aIL-6) MAb ($n = 15$), then challenged with SA on 7 dpv. For SA-Alum control, $n$ are 13. Data were from one to three independent experiments with each data point representing one mouse. The data are presented as mean ± SD of biological replicates. The data in (**a, c, d**) were analyzed by one-way ANOVA with Tukey's posthoc test, while the data in (**b, e**) by Kruskal-Wallis non-parametric one-way ANOVA test. *$p < 0.05$, ***$p < 0.001$. ns, non-significant. dpv, days post-last vaccination. SA, *Staphylococcus aureus*. WT, wild-type. Source data are provided as Source Data File. Mouse image was created by BioRender (Created in BioRender. Hajam, I. (2024) https://BioRender.com/a18v205).

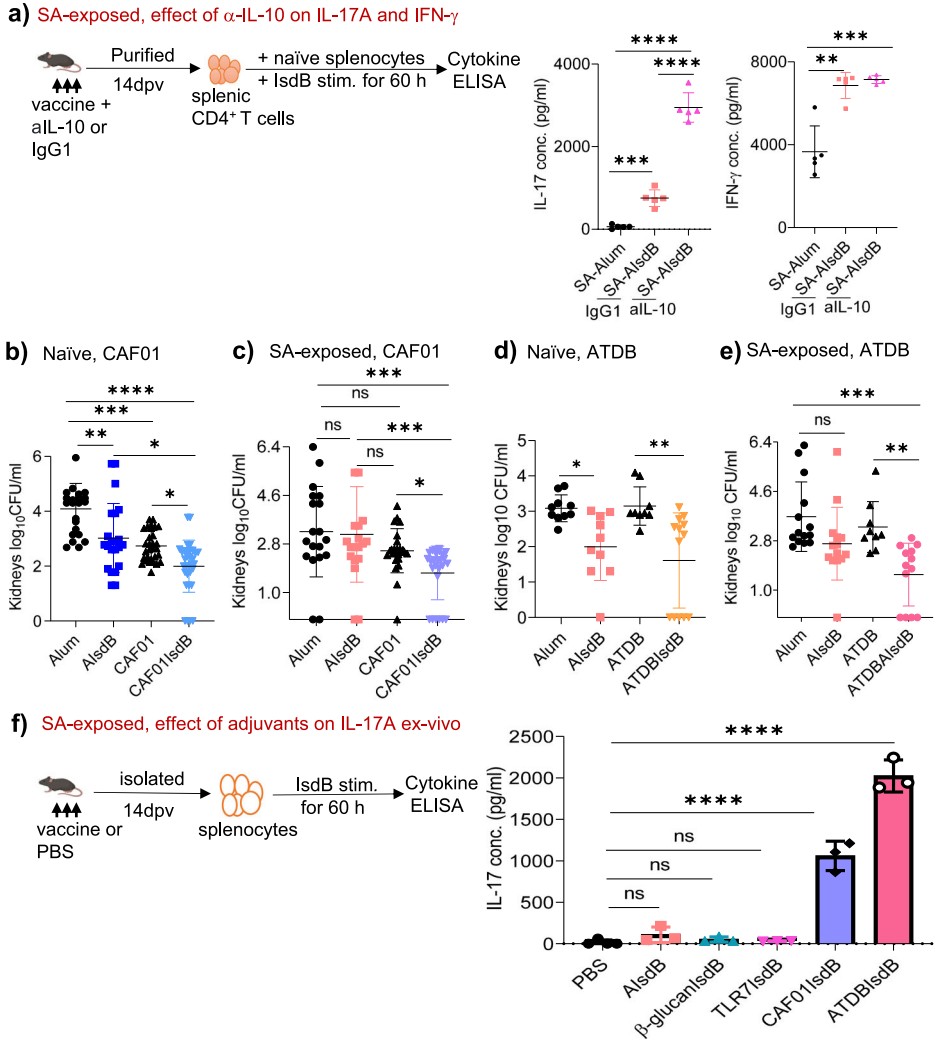

**Fig. 4 | High potency IL-17A promoting adjuvants restore vaccine protection in SA-exposed mice. a** SA-exposed WT mice were treated with either isotype IgG1 control or anti-IL-10 MAb before and on the day of AIsdB vaccination as in Fig.2b. Then, splenic CD4⁺ T cells were purified 14 dpv and incubated with naïve splenocytes plus IsdB (10 µg/ml). IL-17A and IFN-γ from culture supernatant were measured at 60 h ($n = 5$ per group). **b, c** Naïve or SA-exposed WT mice were vaccinated i.p. with Alum ($n = 23$ in **b** and 19 in (**c**)), AIsdB ($n = 22$ in **b** and 20 in c), CAF01 ($n = 30$ in (**b**) and 25 in c) or CAF01IsdB ($n = 32$ in **b** and 26 in **c**), then challenged with SA 14 dpv. **d, e** Naïve or SA-exposed WT mice were vaccinated i.p. with Alum ($n = 10$ in (**d**) and 14 in (**e**)), AIsdB ($n = 10$ in **d** and 14 in e), AlumTDB (ATDB) ($n = 9$ in **d,e**) or ATDBIsdB ($n = 13$ in **d,e**), then challenged with SA 14 dpv. **f** SA-exposed WT mice were vaccinated with either PBS or IsdB plus adjuvant. Splenocytes (pooled from three mouse) were purified 14 dpv and stimulated with IsdB (10 µg/ml) for 60 h, followed by analysis of supernatants for IL-17A. Data were from one to 4 independent experiments with each data point representing one mouse, except in (**f**). The data are presented as mean ± SD of biological replicates, except for (**f**), where data are presented as mean ± SD of three technical replicates of one independent experiment, repeated twice with similar results. The data in (**a, f**) were analyzed by one-way ANOVA with Tukey's post-hoc test while the data in (**b–e**) were analyzed by Kruskal-Wallis non-parametric one-way ANOVA test. $*p < 0.05$, $**p < 0.01$, $***p < 0.001$, $****p < 0.0001$. ns, non-significant. dpv, days post-last vaccination. SA, *Staphylococcus aureus*. WT-wild-type. Source data are provided as Source Data File. Mouse image was created by BioRender (Created in BioRender. Hajam, I. (2024) https://BioRender.com/a18v205).

protective against SA infection, vaccination of ko (-/-) of IL-10Rα on CD4⁺ T cells proved to be protective (Fig. 3c and Supplementary Fig. 4c).

Since IL-10 modulates immune functions through canonical STAT3 dependent signaling, we tested the role of STAT3 using STAT3 inhibitor WP1066, applied before and after vaccination. Consistent with the reported function of IL-10, WP1066 restored AIsdB vaccine efficacy in SA-exposed mice (Fig. 3d). IL-6 also signals through STAT3[29,30], is linked to IL-10 production[31], and is significantly upregulated in SA-exposed mice (Supplementary Fig. 4d). However, blockade of IL-6 at the time of vaccination did not improve AIsdB vaccine efficacy (Fig. 3e and Supplementary Fig. 4e). Overall, we conclude that the suppression of IsdB vaccine after SA exposure is driven by abundant IL-10 by CD4⁺ T cell imprints, and acts principally on IL10Rα on CD4⁺ T cells and STAT3 signaling.

## High potency IL-17A promoting adjuvants restore efficacy of IsdB vaccine in SA-exposed hosts

Numerous studies have underscored the importance of IL-17A and IFN-γ in anti-SA immune responses[32,33]. Our own study demonstrated that blocking of IL-10 enhanced IL-17A in a model of SA reinfection[17]. Therefore, we vaccinated mice with AIsdB in the presence of control or anti-IL-10 (aIL-10) antibodies, then evaluated ex vivo purified CD4⁺ T cell IL-17A and IFN-γ responses to IsdB or HKB (Fig. 4a and Supplementary Fig. 5a). CD4⁺ T cells from SA-exposed IsdB-vaccinated mice produced higher IL-17A responses to IsdB but reduced IL-17A to HKB compared to CD4⁺ T cells from SA-exposed control mice. With either antigenic stimulation, anti-IL-10 treatment of the vaccinated mice significantly augmented IL-17A secretion by the CD4⁺ T cells, without significant effect on IFN-γ responses. The data thus suggests that suppression of SA vaccine by IL-10 is mediated through modulation of IL-17A responses.

To evaluate the significance of reduced IL-17A on loss of vaccine protection, we infected mice with SA, ex vivo stimulated splenocytes with a panel of adjuvants, then measured IL-17A in the supernatants (Supplementary Fig. 5b). Among the tested adjuvants, CAF01 and the combination of synthetic cord factor (TDB) and alum (ATDB) stimulated the most potent IL-17A responses. Next, we tested the adjuvants in combination with IsdB in vivo. In naive mice, all adjuvants plus IsdB reduced bacterial burden in response to SA challenge (Fig. 4b, d, and Supplementary Fig. 6a, c, d, f, h). In comparison, only adjuvants that induced the highest levels of IL-17A (CAF01 and ATDB), in combination with IsdB, protected SA-exposed mice (Fig. 4c, e, and Supplementary Fig. 6b, e), whereas adjuvants that induced lower levels of IL-17A (TDB, β-glucan and TLR7) were not protective (Supplementary Fig. 6c, g, h). Notably, the CAF01 adjuvant alone was protective, and the likely innate immune protection conferred by CAF01 was more pronounced in naive mice than SA-exposed mice (Fig. 5b, c and Supplementary Fig. 6a, b), suggesting that prior SA exposure also lowered CAF01 induced innate immune responses. In addition, we tested pertussis (Ptx) and diphtheria (Dpth) toxoids as adjuvants. We showed that pertussis toxoid-IsdB (Ptx-IsdB) but not diphtheria toxoid-IsdB (Dpth-IsdB) fusion proteins protected SA-exposed mice from SA challenge (Supplementary Fig. 7a, b).

To determine IL-17A levels that are induced by adjuvanted protective or non-protective IsdB vaccinations, we restimulated splenocytes with IsdB or HKB ex vivo (Fig. 4f and Supplementary Fig. 7c). Protective vaccines (IsdB plus CAF01 or ATDB) induced higher IL-17A (Fig. 4f) and TNF (Supplementary Fig. 7c) compared to non-protective IsdB vaccines. In comparison of AIsdB or Ptx-IsdB vaccine groups (Supplementary Fig. 7d), higher IL-17A response was also observed in Ptx-IsdB group. Remarkably, IsdB adjuvanted with TLR7, CAF01, and ATDB also induced the highest levels of IL-10 (and IFN-γ) among IsdB vaccinated groups. Taken together, these findings suggest that adjuvants that induce potent IL-17A responses can overcome antigen-specific vaccine suppression in hosts previously exposed to SA.

### IL-17A and IFN-γ, in spite of abundant IL-10, elicits CAF01IsdB vaccine protection in SA-exposed hosts

Next, we honed in on the mechanisms responsible for CAF01IsdB vaccine protection in mice previously exposed to SA. First, we determined through antibody transfer that CAF01 confers no or modest humoral protection in the tissues of vaccinated naïve or SA-exposed mice in spite of generating abundant specific antibodies (Fig. 5a, and Supplementary Fig. 8a–c). Consistent with CAF01's T cell dependent mechanism of adjuvancy, CAF01IsdB vaccination of SA-$muMt^-$ mice (Fig. 5b and Supplementary Fig. 8d) and adoptive transfer of CD3$^+$ T or CD4$^+$ T cells from SA-exposed CAF01IsdB vaccinated mice, but not CAF01 treated mice, conferred protection against SA challenge in recipients (Fig. 5c, d, and Supplementary Fig. 8e, f). As noted previously, CAF01 alone conferred significant antigen-independent protection, likely from the induction of innate training mechanism. CAF01IsdB conferred additional T cell protection. Like CAF01, the combination of TDB and alum also restored CD4$^+$ T cell dependent protection in SA-exposed host (Supplementary Fig. 8g).

To visualize cytokine expression by CD4$^+$ T cells induced by CAF01IsdB compared to AIsdB in SA-exposed mice, we performed intracellular cytokine staining (ICS) (Supplementary Fig. 8h), with corroboration by ELISA (Supplementary Fig. 9a). CAF01IsdB vaccination induced higher single-positive IFN-γ, IL-17A, and IL-10 expressing CD4$^+$ T cells, in addition to the presence of double-positive IL-17A/IFN-γ or IL-17A/IL-10 CD4$^+$ T cells than AIsdB vaccination in SA-exposed mice (Fig. 5e).

To determine the critical protective role of IFN-γ induced by CAF01IsdB vaccination, we treated CAF01IsdB vaccinated mice with anti-IFN-γ (aIFN-γ) Mab. The treatment induced loss of protection in SA-exposed mice (Fig. 5f, and Supplementary Fig. 9b), indicating the importance of IFN-γ in protection. In comparison, we tested the importance of IL-17 in CAF01IsdB mediated protection by anti-IL17A (aIL-17A) blockade and by the use of IL-17A$^{-/-}$ mice. Notably, protection induced by CAF01 alone was not different in WT and IL17A$^{-/-}$ mice, consistent with proposed innate training. In contrast, antigen-dependent protection was lost in SA-IL-17$^{-/-}$ mice (Fig. 5g and Supplementary Fig. 9c). Mice treated with aIL17A antibody also suggest loss of protection in the kidneys (Supplementary Fig. 9d).

Taken together, our findings suggest that potent IL-17A and IFN-γ promoting adjuvants, in spite of IL10, could overcome vaccine interference mediated by prior SA exposure. Although CAF01 promotes IsdB vaccine protection via IL-17A, CAF01 induced IFNγ is clearly important, if not more important than IL-17A.

### IL-10 induces non-IsdB vaccine interference and cross-suppression of protective vaccines, but is overcome by CAF01 adjuvancy

The demonstration of IL-10 linked imprints as critical suppressor of IsdB vaccination prompted us to query the broader implication of the finding on non-IsdB vaccine failures and on the potential for IL-10 to cross-suppress effective vaccines when protective and non-protective vaccines are combined. We, thus, evaluated another SA antigen, alum adjuvanted vaccine against ferric-hydroxamate uptake protein, FhuD2, which is protective in naive mice, but not in SA-exposed mice[16]. We showed that the application of anti-IL-10 treatment restored protective efficacy of the FhuD2 vaccine in SA-exposed mice (Fig. 6a), consistent with the broader role of IL-10 in vaccine suppression.

Multi-component staphylococcal vaccines are routinely evaluated in clinical trials. We speculated that mixing a protective and non-protective vaccine antigen could lead to cross-suppression of protective vaccine by IL-10 released by the suppressive imprint. Thus, we vaccinated mice with a previously identified protective sdrE antigen (Fig. 6b) in combination with IsdB antigen. Compared to stand-alone protective sdrE vaccine (Supplementary Fig. 10a), the combination of IsdB and sdrE was non-protective in the SA-exposed host (Fig. 6b). But this non-protective phenotype was reversed with blockade of IL-10. Adjuvancy with CAF01 could overcome, albeit moderately, IsdB suppression of combination IsdB/sdrE vaccine in SA-exposed mice (Fig. 6c and Supplementary Fig. 10b). This proof-of-concept study underscores the potential adverse effect that could come from combining multiple SA antigens in vaccine trials.

In addition to IsdB, we asked if potent IL-17A promoting adjuvants, such as CAF01, could reverse the lack of efficacy of other SA vaccines. Thus, to naïve and SA-infected mice, we administered IsdB, mntC, IsdA and FhuD2 antigens adjuvanted with CAF01. In earlier experiments, we challenged the mice with SA 7 dpv (days post-vaccination) and noted significant antigen-independent CAF01 protective effect. To clearly measure antigen-dependent protection as well as assess durability of protection, we challenged the mice either on D56 or D85 after vaccinations. CAF01 adjuvanted SA vaccines mediated efficient protection in both naive and SA-exposed mice (Fig. 6d–f and Supplementary Fig. 10c–e). To determine if CAF01-IsdB is protective against other strains of SA and in non-i.p. models of SA infection, we repeated the vaccination experiments using SA Newman and SA113 strains (Supplementary Fig. 10f, g) and in a soft-tissue model of SA infection (Fig. 6g, h). Across SA strains and in the soft-tissue infection model, CAF01IsdB proved to be effective. In addition to female mice, male mice also demonstrated protective efficacy against SA challenge with CAF01IsdB vaccine (Supplementary Fig. 10h).

Taken together, our findings strongly support a critical role of IL-10 in suppression of SA vaccines and the potential for IL-10-mediated cross-suppression in combination vaccines. Potent IL-17A and IFN-γ promoting adjuvants could prove useful for overcoming IL-10-linked SA immune imprints.

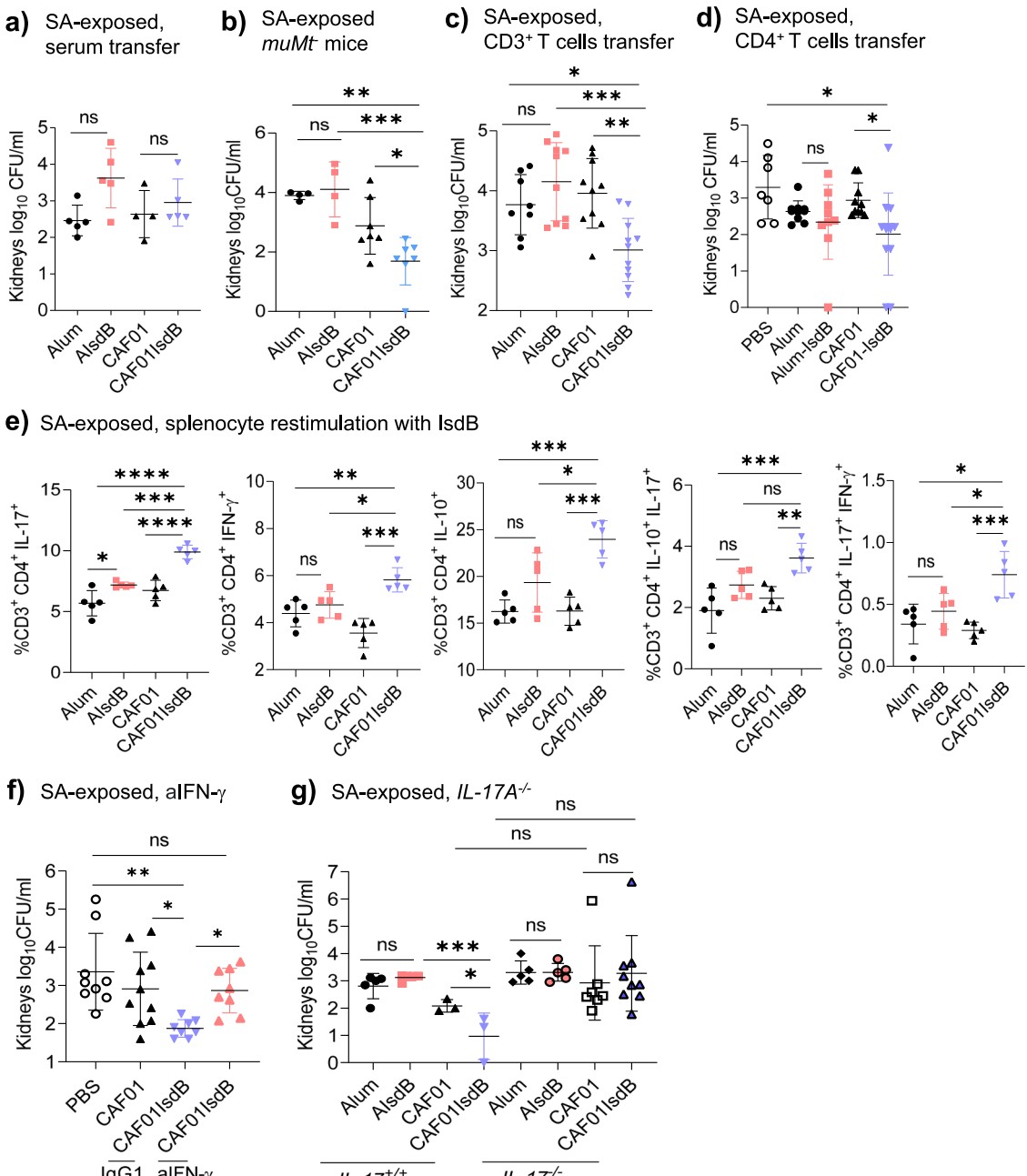

**Fig. 5 | IFN-γ and IL-17A, independent of IL-10, confer IsdBCAF01 vaccine protection against SA in SA-exposed mice. a** Serum (150 µl) collected 14dpv of SA-exposed vaccinated WT mice was transferred i.v. into naive mice. The recipient mice were challenged 20 h after with SA (n = 5 per group). **b** SA-exposed *muMT* mice were vaccinated i.p. with Alum (n = 4), AIsdB (n = 4), CAF01 (n = 7) or CAF01IsdB (n = 6), then challenged with SA 14 dpv. **c, d** SA-exposed WT mice were vaccinated i.p. with PBS (n = 7 in **d**), Alum (n = 8 in **c** and 10 in **d**), AIsdB (n = 10 in **c** and 9 in (**d**), CAF01(n = 10 in (**c, d**) or CAF01IsdB (n = 11 in **c** and 13 in **d**). 14 dpv, splenic CD3⁺ T cells or CD4⁺ T cells were transferred i.v. into naïve WT mice. After 20 h, the recipient mice were challenged with SA challenged. **e** SA-exposed WT mice were vaccinated i.p. with Alum, AIsdB, CAF01 or CAF01IsdB. 14 dpv, splenocytes (1.5 × 10⁶, n = 5) were isolated and restimulated with IsdB antigen (10 µg/ml)

for 48 h, followed by intracellular cytokine analysis by flow cytometry. **f** SA-exposed WT mice were vaccinated i.p. with PBS (n = 9), CAF01 (n = 10) or CAF01IsdB (n = 8). After 30 dpv, the mice were treated with an isotype IgG1 control or anti-IFN-γ (aIFN-γ) Mab, one day before and on the day of SA challenge. **g** SA-exposed WT or IL-17⁻/⁻ mice were vaccinated i.p. with Alum (n = 5), AIsdB (n = 5), CAF01 (n = 3 or 7) or CAF01IsdB (n = 3 or 9), then SA challenged 14 dpv. Data were from one to two independent experiments with each data point representing one mouse. The data are presented as mean ± SD of biological replicates. The data in (**a, b, e, g**) were analyzed by one-way ANOVA with Tukey's post-hoc test, while the data in (**c, d, f**) by Kruskal-Wallis non-parametric one-way ANOVA test *$p < 0.05$, **$p < 0.01$, ****$p < 0.001$, ***$p < 0.0001$. ns, non-significant. dpv, days post-last vaccination. SA, *Staphylococcus aureus*. WT-wild-type Source data are provided as Source Data File.

## Discussion

As a successful human pathobiont, SA depends on well-orchestrated immune evasive strategies to maintain coexistence with the host. Adaptive immune evasive strategies include direct blunting of T and B cell responses directed against the pathogen[34]. We have previously shown that SA exposure induces the development of non-host

protective CD4⁺ T cell responses that had no impact on SA burden[17]. In the current study, we show that the expression of abundant IL-10 by these CD4⁺ T cells allow the pathobiont to suppress anti-SA vaccine responses. The SA-induced suppressive mechanism appears antigen-specific, and is CD4⁺ T cell IL10Rα as well as STAT3- dependent. IL-10 is commonly induced by commensals[35], and the role of IL-10 in SA

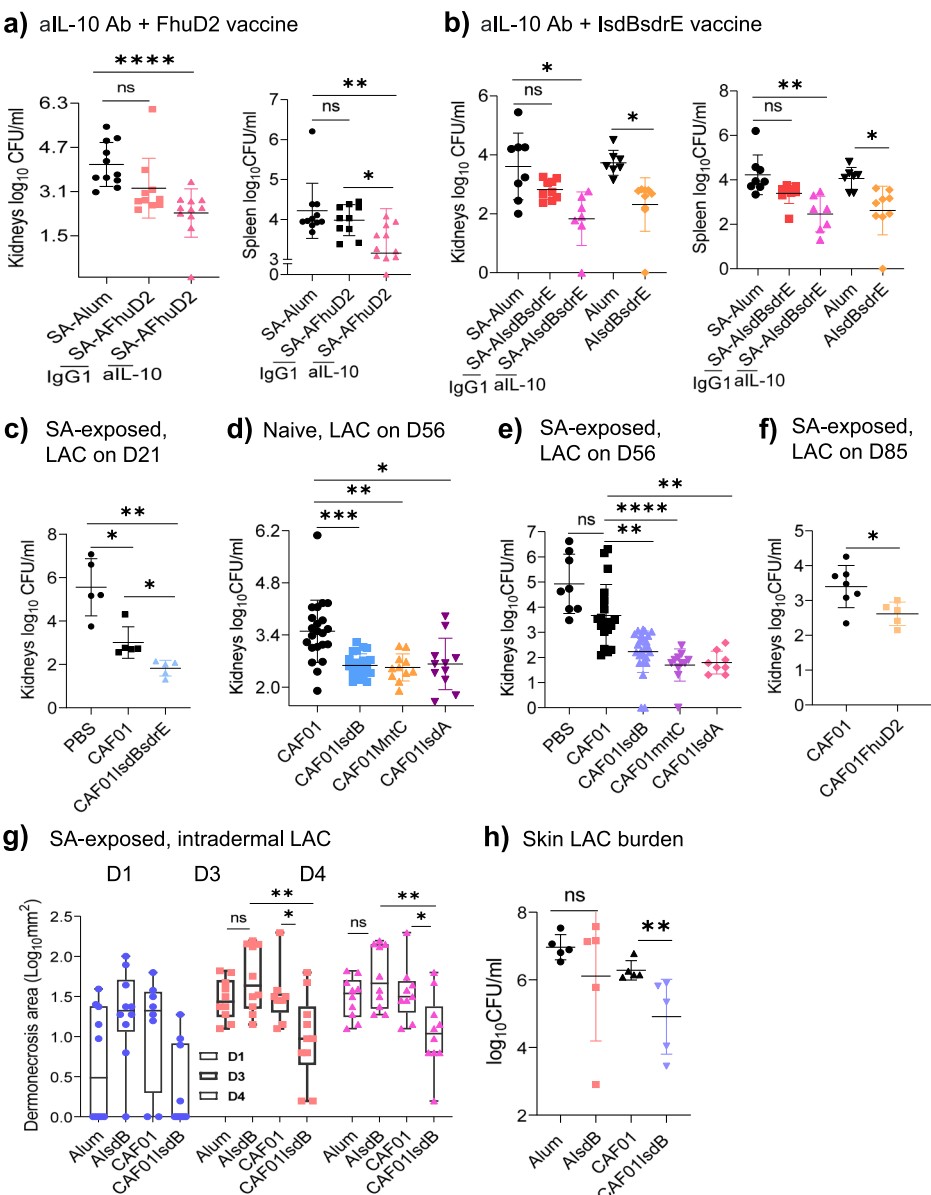

**Fig. 6 | IL-10 mediates suppression and cross-suppression of SA vaccines in SA-exposed mice, but interference can be overcome by CAF01. a** SA-exposed WT mice were treated with either isotype IgG1 Ab or anti-IL-10 MAb one day before and on the day of AFhuD2 vaccination (3x, weekly apart). The mice were SA challenged on 7dpv (*n* = 10). **b** SA-exposed WT mice were treated with either isotype IgG1 Ab (*n* = 9) or anti-IL-10 (aIL-10) MAb (*n* = 7) one day before and on the day of AIsdBsdrE combination vaccine (3x, weekly apart). The mice were SA challenged on 7dpv. For SA-Alum, Alum and AIsdBsdrE, *n* = 8, 7 and 9, respectively. **c** SA-exposed WT mice were vaccinated i.p. with CAF01 or CAF01IsdBsdrE, then challenged with SA 21 dpv (*n* = 5 per group). **d, e** Naïve or SA-exposed WT mice were vaccinated i.p. with PBS (*n* = 8), CAF01 (*n* = 22 in **d** and 19 in **e**), CAF01IsdB (*n* = 23 in **d** and 25 in **e**), CAF01mntC (*n* = 11 in **d, e**) or CAF01IsdA (*n* = 11 in (**d**) and 8 in **e**), then SA challenged on 56 dpv. **f** SA-exposed WT mice were vaccinated with CAF01 (*n* = 7) or

CAF01FhuD2 (*n* = 5), then SA challenged 85 dpv. **g, h** WT mice were exposed to SA i.p., then subcutaneously vaccinated with Alum, AIsdB, CAF01 or CAF01IsdB (3x, weekly apart), then challenged with SA intradermal 7dpv. Skin lesions (**g**, *n* = 10) and CFU measured on 4 days after (**h**, *n* = 5). Data were from one to four independent experiments with each data point representing one mouse. The data are presented as mean ± SD of biological replicates, except for (**g**). The data in (**a, b, d, e, g**) were analyzed by Kruskal-Wallis non-parametric one-way ANOVA test, data in (**c**) by one-way ANOVA test, while the data in (**f, h**) by two-tailed non-parametric unpaired Mann-Whitney T test. Box plots in (**g**) are presented as min. to max. *p < 0.05, **p < 0.01, ***p < 0.001, ****p < 0.0001. ns, non-significant. dpv, days post-last vaccination. SA, *Staphylococcus aureus*. WT-wild-type. Source data are provided as Source Data File.

immunobiology is becoming increasingly appreciated. The induction of IL-10 has been reported in association with PSMs, SpA and staphylococcal TSST[21,22,36]. We have previously shown that O-acetylation of cell wall peptidoglycan suppresses IL-1β, IL6, IL-22, and TGF-β, resulting in muted Th17 development and increased IL-10[17]. During initial colonization, IL-10 is produced primarily by myeloid cells which facilitate coexistence with the host through impairment of T cell responses[28]. Relatedly, a biofilm model shows that IL-10 secreted by

MDSCs mediates histone deacetylase complex-dependent regulation of biofilms[37]. Upon SA infection, IL-10 is abundantly produced and can drive host protection or pathology depending on the site of infection[38]. In a human study of staphylococcal invasive diseases, Sakoulas and colleagues showed that IL-10 is associated with greater mortality, which correlates strongly with higher SA burden[39]. Although this result could be interpreted in many ways, it could be consistent with increased IL-10 in the context of SA infection that facilitates

greater SA survival. Investigating the cellular source of IL-10 in these patients could provide more insight.

The source of IL-10 can be both myeloid cells[28,40] or T cells[17,22]. In our study, amplified CD4+ T cells are the principal source of IL-10 in response to antigen-specific stimuli. Production of IL-10 by CD4+ T cells is the primary driver of SA vaccine suppression, and appears to be more important than IL10 produced by myeloid cells. IL-10 suppression of vaccines appears to occur through downregulation of the major anti-SA IL-17 axis[41]. Notably, IL-10 is critically important for the generation of CD4+ Tregs via IL-10Rα and STAT3 signaling[41]. In the absence of STAT3, T cells do not express IL-10, and similar to the IL-10r-deficient Treg cells, lose their ability to suppress Th17 responses and drive Th17-driven colitis[41,42]. Furthermore, pharmacological blockade of STAT3 signaling also reduced Tregs generation, thus permitting the secretion of increased host-protective T-cell effector cytokines[43].

Two prior studies[22,44] have demonstrated that the application of IL-10 neutralizing antibodies could enhance or reverse anti-SA vaccine efficacy. In the study by Narita and colleagues, an anti-superantigen Toxic shock syndrome toxin 1 vaccine effectively protected naïve mice from SA infection via a Th17 mediated mechanism. However, protection was lost after 12 weeks but could be restored with anti-IL-10 antibody treatment. The more recent study by McLoughlin's group showed that the application of anti-IL10 at the time of SA Clumping factor A (ClfA) vaccination with T-cell adjuvant CPG enhanced IL-17 and IFN-γ production and protection against SA. These findings support the inhibitory role of IL-10 as well as targeting of IL-10 in SA vaccination. In our study, the induction of IL-17 (and IFN-γ) by various adjuvants reversed IsdB vaccine efficacy. Unexpectedly, both CAF01 and TDB-Alum elicited higher IL-10 expression in T cells, suggesting that the absence of IL-17, rather than presence of IL-10, is responsible for lack of protection in AIsdB vaccinated SA-exposed mice. Our study revealed that protective T cells elicited by CAF01 secrete more than one cytokine, IL-17A/IL-10 or IFN-γ/IL-17A, and blockade of either IL-17 or IFN-γ abrogated vaccine protection conferred by CAF01. While IFN-γ/IL-17A CD4+ T cells have been associated with improved antimicrobial functions, abundant IL-17 and IFN-γ secretion could lead to excessive or dysregulated inflammation and tissue damage rather than disease resolution[45,46]. Thus, co-expression of IL-10 by these CD4+ T cells could mitigate against tissue damage. The emerging understanding of Treg cells adds a layer of complexity and opportunity to this discussion. Tregs are known for their regulatory function in immune responses, helping to maintain immune homeostasis and tempering excessive inflammation. Recent findings suggest that Tregs can play a significant role in modulating IL-17 responses during infections[47]. This regulatory function positions Tregs as potential targets for therapeutic interventions aimed at balancing immune responses to achieve effective pathogen clearance while minimizing collateral tissue damage.

Most current SA vaccines in clinical trials consist of combination vaccines, likely reflecting an economic strategy from past vaccine failures. We show that when a protective vaccine is combined with a suppressive vaccine, cross-suppression mediated by IL-10 could lead to the loss of overall vaccine efficacy. Conversely, IL-10 suppression of IL-17A expression prompted the study of IL-17A promoting adjuvants. Of these, only those adjuvants that induced the highest levels of IL-17A (CAF01, ATDB and pertussis toxoid) were protective in both naïve and SA-experienced hosts and demonstrated durable protection. CAF01 in particular reversed all blunted staphylococcal CWA antigen vaccines we tested, by a mechanism that is both IL-17A and IFN-γ dependent. We did not test the efficacy of CAF01 with SA toxin vaccines since anti-toxin imprints appear protective in our i.p. infection model[48]. However, Teymournejad and colleagues demonstrated suboptimal anti-toxin vaccine responses when mice are previously infected in the skin with SA. CAF01 adjuvancy in that model restored anti-toxin protection against SA skin challenged[20]. The proposed mechanism of vaccine interference by toxin imprints in the skin is the depletion of DCs that serve as APCs. How CAF01 boosting of IL-17A and IFN-γ led to reversal of vaccine efficacy is not clear. The role of IL10 in the suppression of toxin vaccines in the skin is also not clear and merits investigation. Although CAF01 is a highly promising adjuvant that overcomes IL-10 mediated suppression, it is notable that correlates of immunity in humans have been less clearly defined, and IFN-γ appears to be the crucial factor in macrophage-mediated control of infection humans[49,50].

Our current study sheds light on how vaccination against pathobionts could be complicated by microbial coexistence strategies. Hence, the study could provide important insight on the particular difficulty associated with vaccinating against pathobionts and overcoming these difficulties. The proposed immune imprinting mechanism of pathobionts bears similarity to the Original antigenic sin hypothesis which posits that initial immune responses to influenza are preferentially recalled on subsequent infection or vaccination with a variant influenza strain[51,52]. However, unlike influenza, imprints generated by pathobiont SA are largely non-protective or suppressive which permits microbial coexistence with the host. This difference has important ramifications for SA vaccinology. Consistent with our data supportive of a vaccine suppression mechanism, there was a five-fold greater mortality associated IsdB vaccination compared to placebo in the IsdB vaccine trial, with report of lower number of Th17 cells among those who succumbed[15,53]. Measurement of IL-10 in that study would have provided further insight on the reason for greater mortality.

SA is one of a group of pathobionts, ESK(C)APE organisms, that pose significant threat to public health because of their acquisition of antibiotic resistance[2], and hence are major vaccine targets. Like SA, vaccine failures have been common among this group. While vaccine against Klebsiella has presented a challenge because of capsule and LPS diversity[54], why vaccinations against other pathogens such as C. difficile and P. aeruginosa have been repeatedly unsuccessful is unclear[55,56]. Notably, among commensals, microbial induction of IL-10 is common[35]. Among hard-to generate vaccines, C. difficile and pathogens like M. tuberculosis, Herpes Simplex Virus, Human Immunodeficiency Virus, and plasmodium species are also associated with abundant IL-10 per reports[57-60]. The role of IL-10 in circumventing vaccine protection against these pathogens would be worth exploring.

Although pathobiont interference with vaccination is unlikely to be an evolutionary trait, we propose that the mechanism is likely advantageous to SA as human sera routinely harbor high levels of non-protective anti-SA imprints, otherwise non-beneficial to the host[61,62]. In addition to permitting reinfection, we speculate that these imprints could support host-SA coexistence by two mechanisms. First, SA, like most commensal microbes, expresses cell surface antigens with significant homology to antigens expressed by other pathogens, for instance, mntC[63], IsdB NEAT domains[64], and Als3p[65]. Infection of the common host by pathogens expressing homologous surface antigens risks eliciting cross-reactive T cells that could oust SA. As an example, Weiser and colleagues have shown that cross-reactive antibodies to a S. pneumoniae dehydrogenase inhibited subsequent colonization by SA[66]. We hypothesize that the existence of CD4+ T cell imprints that sense cross-reactive antigens and expresses IL-10 would represent an effective way to suppress such potential threats. An additional benefit of linking IL-10 to anti-SA immune imprints is the potential to deliver abundant IL-10 to promote SA survival at sites of colonization or infection, through IL-10's broad innate immune suppressive functions[67,68]. In Mycobacterium tuberculosis, the presence of IL-10 rich environment established by CD4+ T cells promotes survival of the pathogen[69]. Suppressor CD4+ T cells are required for the maintenance of persistent Leishmaniasis infection in a mouse model[25], and the absence of IL-10 supports the development of spontaneous murine enterocolitis from fulminant proinflammatory T cell responses to normal bacterial flora[70]. Hence, we propose that by coupling IL-10 to

SA-specific CD4[+] T cells, SA can effectively suppress various immune mechanisms potentially harmful to SA including iatrogenic SA vaccines.

Our study has a number of limitations. While our model can explain the lack of success of staphylococcal vaccine trials, it is unclear how closely the model mimics the human immune environment of a previously colonized or infected individual. Our specific model of 3 successive PI infections may not be generalized to natural SA exposure, although we have published additional models of SA-preexposure that used only one IV SA infection followed by vancomycin treatment, 2 i.p. SA infection, and two soft tissue infections. Each of the models demonstrated robust suppression of IsdB vaccination following the SA-exposure. Irrespective, until a SA vaccine trial is successful, it may be difficult to validate the full impact of immune imprinting on human vaccination. Pathogen aggregation by antibodies has the potential to drive a deleterious response to the host. For example, IgGs generated against enterococcal aggregation substance can enhance *Enterococcus faecalis* aggregation and increase the severity of infective endocarditis[71]. Thus, vaccination against SA cell surface molecules, which mainly generate non-protective antibody responses[48], has the potential to promote microbial aggregation and contribute to more severe SA infections. Our study uses primarily bacterial burden for readouts and could be enhanced by evaluation of pathology. Harvest of bacteria at 20 h post SA infection may not be the optimal timing for the assessment of protective T cell effect. Furthermore, unlike experiments where SA challenge is carried out beyond 50 dpv, results of experiments where SA challenge is measured 7-14 dpv may partly reflect the function of innate immune cells or effector memory T cells in circulation rather than antigen-recall responses. Our study does not exclude the possible role for innate training in protective IsdB/alum T cell response, although adoptive T cell transfer experiments indicate that T cells are certainly important for protection and suppression. For some experiments, we have noted CFU differences in one organ and not the other. This likely reflects the smaller magnitude of T cell responses compared to whole T and B cell responses to IsdB vaccine of our prior studies[16,48]. Overall, the large number of approaches we have taken to characterize the phenotype was to assure that our conclusions are largely robust in spite of the smaller magnitude of T cell findings.

In summary, we presented a pathobiont mechanism of adaptive immune suppression that leaves enduring imprints within the host immune system. We show that the mechanism makes human effort to vaccinate against the bacterium difficult. As pathobionts become increasingly targets of human vaccine because of threat of antibiotic resistance, vaccine development strategies need to consider these pathogen strategies.

## Methods

### Mouse strains
C57BL/6 wild-type (Strain000664), B6.129S2-*Ighm*[tm1Cgn]/J (Strain: 002288, muMt⁻), B6.Cg-Il17a/Il17f[tm1.1Impr] Thy1a/J (Strain: 033431), B6.Cg-Tg(CD4-cre)1Cwi/BfluJ (Strain: 022071), B6.129P2-Lyz2tm1(cre) Ifo/J (Strain: 004781), B6(Cg)-Il10[tm1.1]Karp/J (Strain: 014530), B6(SJL)-Il10ra[tm1.1Tlg]/J (Strain: 028146), and B6.129P2(C)-Il10[tm1Roer]/MbogJ (Strain: 036598) were purchased from Jackson laboratories. Mice were housed and bred in ventilated cages under specific-pathogen-free conditions, constant ambient temperatures (21–25 °C) and humidity (50-60%), and with standard 12-h light/12-h dark cycles. Six- to 8-week-old female mice were used for all animal experiments except for the experiment in Supplementary Fig. 10h where male mice were used. All animal studies were approved under the guidelines of the University of California San Diego (UCSD) Institutional Animal Care and Use Committee. Mice were housed in an animal facility at UCSD with a standard of care as per federal, state, local, and NIH guidelines.

### Bacterial strains and growth conditions
Frozen stocks of SA Becker, SA Becker IsdB/HarA deletion mutant (gifts from Dr. Secore, Merck)[72], SA USA300 (LAC)[73], SA Newman and SA113 (courtesy of Dr. Fritz Goetz)[74] in 50% glycerol were streaked on sheep blood agar plates and incubated overnight at 37 °C. Then a single colony of SA was routinely cultured in Todd Hewitt broth (THB; SKU 4983962, Neogen, USA) and an overnight bacterial culture was diluted 1:200 in THB and grown at 37 °C under continuous shaking (200 rpm) until an optical density (OD) of 0.75–0.9. Bacteria were washed twice with sterile phosphate buffer saline (PBS; 21-040-CV, Corning, USA; pH 7.4) prior to use. The inocula were confirmed by determination of colony forming units (CFU) on agar plates.

### Gene cloning and protein expression
The *IsdB* gene without signal peptide was amplified from the genomic DNA of LAC USA300 using primers: 5'IsdB (5′- TAAGGCCTCTGTC-GACGCAGCTGAAGAAACAGGTGGT-3′) and 3'IsdB (5′-CAGAATTCG-CAAGCTTTAGTTTTTTACGTTTTCTAGGT-3′). The PCR product was cloned into pET6xHN-N expression vector (In-Fusion Ready, 631433, Takara Bio, USA) and the resultant recombinant plasmid (pET-IsdB) was transformed into *Escherichia coli* BL21 (DE3) pLysS strain (69451-4, Novagen, USA) for protein expression. Overnight cultures of recombinant *E. coli* were diluted 1:100 into LB broth containing 100 µg of ampicillin/ml (A9518, Millipore Sigma, USA) and the bacterial cultures were allowed to grow at 37°C under continuous shaking (200 rpm) until an OD of 0.4. Then IsdB protein expression was induced by the addition of 1 mM isopropyl-β-D-thiogalactoside (IPTG; 16758, Millipore Sigma, USA) and 2 h post induction, the bacterial pellet was collected and resuspended in lysis buffer ((50 mM $Na_2HPO_4$ (S9763, Honeywell Fluka, USA) (pH 7.4), 300 mM NaCl (18-214, Apex Bioresearch, USA), 2 mM $MgCl_2$ (M8266, Sigma-Aldrich, USA), 10 mM imidazole (12399, Sigma-Aldrich, USA), 0.1% Tween 80 (P4780, Sigma-Aldrich, USA), 1% Triton X-100 (X100, Sigma-Aldrich, USA), 1 mg/ml egg white lysozyme (J60701-14, ThermoScientific, USA), 1 mM PMSF (P-470-25, Gold Biotechnology, USA), and 10 µg/ml DNases (EN0521, ThermoScientific, USA); pH-7.4)). His-tagged IsdB protein was purified from the clarified lysate by incubation with the His60 Ni Superflow resin (635660, Takara Bio USA, Inc.) for 2 h, followed by passing the mixture through chromatographic gravity columns (786-197, G Biosciences, USA). The column was washed thrice with wash buffer (50 mM $Na_2HPO_4$ (pH 7.4), 300 mM NaCl and 35 mM imidazole; pH-7.4), followed by elution with elution buffer (10 mM $Na_2HPO_4$ (pH 7.4), 300 mM NaCl, 300 mM imidazole, 0.1% Tween 80; pH-7.4). The eluted protein was washed thrice with PBS-T buffer containing 0.1% tween-80 using 50 kDa Amicon centrifugal filters (UFC905024, Millipore Sigma, USA). The purity of purified proteins was confirmed by SDS-PAGE analysis, and the protein was clarified from LPS contamination using Pierce high-capacity endotoxin removal spin columns (88274, ThermoFisher Scientific, USA).

To produce pertussis toxoid (Ptx) and diphtheria toxoid (Dpth) proteins, pET28a-Ptx and pET28a-Dpth plasmids were synthesized (GenScript, USA), and transformed into *E. coli* BL21 (DE3) pLysS strain (69451-4, Novagen, USA) for protein expression and purification as discussed above.

To generate Ptx-IsdB and Dpth-IsdB fusion proteins, IsdB gene was cloned into pET28a-Ptx or pET28a-Dpth plasmid at SalI and Xho1 site with a protein bridge (GGGGSGGGGSGGGGS) in between Ptx or Dpth and IsdB protein. The fusion recombinant plasmid (Ptx-IsdB or Dpth-IsdB) was transformed into *E. coli BL21* (DE3) pLysS strain (69451-4, Novagen, USA) for protein expression and purification as mentioned above. FhuD2, IsdA, mntC, and sdrE proteins were purified under native conditions as discussed in our previous report[16]. Ptx protein was purified under denaturing condition as described previous[75].

## Adjuvants and vaccine formulations

Adjuvants Aluminum hydroxide gel (Alum; vac-alu-250, InvivoGen), Gardiquimod (TLR7 agonist; SML0877, MilliporeSigma, USA), β-Glucan from *Saccharomyces cerevisiae* (346210, MilliporeSigma, USA), D-(+)-trehalose 6,6'-dibehenate (TDB; 890808 P, Avanti Polar lipids, USA) and dimethyldioctadecylammonium bromide (DDA; D2779, Sigma-Aldrich, USA) were used. TLR7 agonist was dissolved in DMSO solvent (5 mg/ml; D2650, Sigma-Aldrich, USA) and stored in -20ºC. β-Glucan was dissolved in sterile PBS (pH 7.4) and stored in -20ºC.

25 mg of TDB was dissolved in 500 µl of DMSO, vortexed and then heated at 56 ºC for 30 to 60 s, followed by immediate addition of 24.5 ml of sterile molecular grade water (46-000-CV, Corning, USA). The solution was thoroughly mixed, passed through a fine needle several times, aliquoted and stored in -20ºC.

DDA was dissolved in 0.01 M Tris-HCl solution (816124, MP Biomedicals, USA) and heated at 80 ºC for 20 min by gentle shaking several times. CAF01 adjuvant was prepared by mixing DDA solution and TDB at 5:1 molecular ratio and stored at 4 ºC. Alum (500 µg/mouse), TLR7 agonist (10 µg/mouse), TDB (50 µg/mouse), β-Glucan (200 µg/mouse), or CAF01 (250 µg DAA plus 50 µg TDB) were mixed with the IsdB antigen and the vaccine was placed on ice for 1 h under continuous rocking before mouse vaccination. Desired volume of the vaccine (200 µl/mouse) was made by PBS. TDB (40 µg/mouse) and Alum (500 µg/mouse) were mixed with the IsdB antigen and placed on ice for 1 h under continuous rocking before mouse vaccination. IsdB antigen was used at 70 µg for the first immunization, and 50 µg for the second and third immunizations.

mntC, IsdA, and FhuD2 vaccines were prepared as per IsdB vaccine.

## Prior SA infections, vaccinations, and SA challenge

Prior SA infection in 6-week-old female mice was established by inoculating the mice i.p. with SA ($3 \times 10^7$CFU) three times at 7-D (day) intervals (Fig. 1).

IsdB vaccination consisted of three i.p. injections of IsdB antigen (70 µg, 50 µg and 50 µg) with adjuvant or adjuvant alone at 7-D intervals (main Fig.1). The mice were challenged i.p. 7, 14, 60 or 85 days after the last vaccination with SA ($3 \times 10^7$CFU) and spleen and kidneys were harvested 20 h after. Unless otherwise stated, mice were challenged i.p. 7 dpv (days post vaccination) and tissues were harvested 20 h after. The tissues were homogenized in sterile PBS (500 µl) and then serially diluted with PBS (from $10^{-1}$ to $10^{-3}$) and plated onto THB agar plates. After 24 h of culture, bacterial colonies were counted.

Vaccination with other SA antigens follow the same protocol as IsdB vaccination. Vaccination with the two components IsdB plus sdrE vaccine used 30 µg of each antigen along with either 500 µg of the Alum adjuvant or CAF01 adjuvant.

For soft tissue infection study, mice were infected i.p. with SA ($3 \times 10^7$) as before, then vaccinated subcutaneously with IsdB antigen plus adjuvant, and intradermally challenged with SA 7 days after the last vaccination. Skin lesions were monitored daily for 4 days and then aseptically excised for CFU enumeration.

To harvest mouse organs and tissues, mice were humanely euthanized using a $CO_2$ gas chamber. In all cases, if a mouse lost 20% of its body weight, had labored breathing, was unable to reach food or water, it was promptly and humanely euthanized using $CO_2$ inhalation.

## Adoptive transfer of CD3⁺ T and CD4⁺ T cells

Spleens were collected 7 days after the last SA infection or 14 days after the last vaccination. The splenocytes were homogenized in sterile PBS (pH 7.4), RBC were lysed (00-4300-54, eBioscience, USA) and CD3⁺ T cells (19851, Stem Cell Technologies, USA), CD4⁺ T cells (19852, Stem Cell Technologies, USA) and B cells (19844, Stem Cell Technologies,

USA) were isolated by negative selection using Stem Cell Technologies EasySep Magnetic Cell Separation kits as per the manufacturer's instructions. The purity of cell populations was greater than 95%. Naïve recipient mice were administered $10 \times 10^6$ CD3⁺ T cells or $5–10 \times 10^6$ CD4⁺ T cells by retro-orbital injection performed under Isoflurane (501017, Fluriso, Vet One, USA) anesthesia. Twenty hours after cell transfer, the mice were challenged i.p. with SA ($2 \times 10^7$CFU), and spleen and kidneys were collected for CFU determination as described above.

In some studies, mice administered CD4⁺ T cells IV and 20 h later, they were immunized 3 times with weekly Alum or Alum plus IsdB vaccine, then challenged with SA ($3 \times 10^7$CFU) on D7 post-last vaccination for CFU determination.

## Ex vivo restimulation of splenocytes and T cells

Splenocytes ($1 \times 10^6$) isolated on 14dpv after the last immunization were left unstimulated or stimulated ex vivo with either an IsdB antigen (10 µg/ml) or a heat-killed SA bacterium (HKB; 1:10) for 2 days. The cells were plated in 96 well cell culture plates (353072, corning incorporated, USA) and cultured in complete RPMI medium (R8758, Sigma, USA) supplemented with 10% fetal bovine serum (FBS; S11150, Atlanta Biologicals, USA) and 1x penicillin–streptomycin antibiotics solution (P4333, Sigma-Aldrich, USA). After 2 days of incubation in a humidified $CO_2$ incubator at 37 ºC, the cells were centrifuged at 450×g and culture supernatants were analyzed for cytokines by a solid-phase sandwich enzyme-linked immunosorbent assay (see cytokine ELISA below).

Ex vivo CD4⁺ T cell restimulation were performed as per splenocyte assay. Purified CD4⁺ T cells ($5 \times 10^5$) were isolated 14 days after the last vaccination, added to freshly isolated naïve splenocytes ($1.5 \times 10^6$) plus IsdB antigen (10 µg/ml) or HKB (1:10), and incubated for 2 days, then analyzed for cytokines as described above.

For detection of intracellular cytokines, splenocytes ($1.5 \times 10^6$) were stimulated with an IsdB antigen (10 µg/ml) for 30 h, then with added brefeldin A (B7651, Sigma-Aldrich, USA) for another 12 h before analysis by intracellular cytokine staining and flow cytometry.

## Depletion of a specific-cell type and specific-cytokines

For depletion of T cell subsets (CD4⁺ T, CD8⁺ T or γδ T cells), the day before and on the day of SA challenge, vaccinated mice were treated i.p. with 300 µg of an *InVivoMAb* anti-mouse CD4 (BE0003-1, BioXCell, USA), *InVivoMAb* anti-mouse CD8α (BE0061, BioXCell, USA), *InVivoMAb anti-mouse* TCR γ/δ (BE0070, BioXCell, USA) or isotype anti-mouse IgG control (10400 C. ThermoFisher Scientific, USA).

For depletion of CD25 and CD69 type of T cells, the day before vaccination, mice were treated i.v. with a single injection of 250 µg of an *InVivoMAb* anti-mouse CD25 (IL-2Rα) (BE0012, BioXCell, USA), *InVivoMAb* anti-mouse CD69 (BE0330, BioXCell, USA) or *InVivoMAb* mouse IgG1 isotype control (BE0083, BioXCell, USA). Following antibody treatment, the mice were vaccinated 3 times at 7-D intervals and then challenged with SA as described.

For anti-IL-6 or anti-IL-10 treatment, 25 µg of an *InVivoMAb* anti-mouse IL-6 (BE0046, BioXCell, USA), *InVivoMAb* anti-mouse IL-10 (BE0049, BioXCell, USA) or *InVivoMAb* rat IgG1 isotype control (BE0088, BioXCell, USA) antibody was given i.v. the day before and on the day of vaccination. The treatment was repeated three times as described in Fig. 2.

For depletion of IFN-γ, the day before and on the day of SA challenge, vaccinated mice were treated i.p. with 100 µg of an *InVivoMAb* anti-mouse IFN-γ (BE0054, BioXCell, USA) or rat IgG1 isotype control (BE0088, BioXCell, USA).

For depletion of IL-17A, the day before and on the day of SA challenge, vaccinated mice were treated i.p. with 100 µg of an *InVivoMAb* anti-mouse IL-17A (BE0173, BioXCell, USA) or mouse IgG1 isotype control (BE0083, BioXCell, USA).

## STAT3 Inhibitor treatment

SA-exposed mice were treated i.p. with 25 μM of STAT3 Inhibitor III (WP1066; 573097, Millipore Sigma, USA) starting the day before vaccination and continued for another two days. The mice were similarly treated with WP1066 for the second and third immunizations. Control vaccinated mice received vehicle alone. Seven days after the last vaccination, mice were challenged with SA, and organs were collected for CFU determination as mentioned above. WP1066 was dissolved in DMSO at 5 mg/ml concentration and stored in -20 before use. Before treatment, 5 μl (25 μM) of WP1066 or 5 μl of DMSO was mixed with 195 μl of sterile PBS and delivered i.p. to mice.

## Serum transfer

Sera (150 μl) collected from adjuvant control or IsdB immunized mice on D7 post-last vaccination was diluted with 50 μl of PBS and injected i.p. Mice were challenged 4 h later with SA ($2 \times 10^7$ CFU). Twenty hours post-challenge, spleen and kidneys were collected for CFU enumeration.

## Serum antibody ELISA

IsdB-specific antibody titers in vaccinated sera, collected on D7 post-last vaccination, were measured by an indirect ELISA as described previously[16]. Briefly, sera were 10-fold serially diluted in PBS-T buffer (9997S, Cell Signaling Technology, USA) containing 1% bovine serum albumin (BSA; A2153, Sigma-Aldrich, USA) and 100 μl was added to 96-well high-binding ELISA plates (655081, Greiner BIO-ONE, USA) pre-coated with recombinant IsdB antigen (1 μg/well). Antigen-antibody reaction was detected by horseradish peroxidase (HRP)-conjugated goat anti-mouse IgG (405306, Biolegend, USA). The plates were developed using TMB substrate (555214, BD OptEIA, USA) and read at optical density (OD) of 450 nm in a multimode microplate reader (PerkinElmer, Waltham, MA, USA).

## Cytokine ELISA

IL-17A (432501, Biolegend, USA), IFN-γ (430801, Biolegend, USA), IL-10 (431414, Biolegend, USA), and TNF (430901, Biolegend, USA) from culture supernatants of ex vivo stimulated cells were measured by a solid-phase sandwich ELISA using commercially available cytokine ELISA kits as per the manufacturer's instructions. 50 μl of the culture supernatant diluted with equal volume of PBS-T buffer (containing 1% BSA) was used for detection of a particular cytokine. The assays were performed in at least triplicates as noted in the figure legends. Cytokine standards (provided with the kits) were run alongside samples to determine cytokine concentrations. Plates were developed and the optical density (OD) was read at 450 nm with wavelength correction set at 570 nm, utilizing a multimode microplate reader (PerkinElmer, Waltham, MA, USA).

## ELISPOT assay

IL-10 secreting B and CD4$^+$ T cells were enumerated using the mouse IL-10 ELISPOT kit (EL417, Bio-Techne, Minneapolis, USA) as per the manufacturer's instructions. Briefly, splenic purified B ($1 \times 10^5$, $n = 5$) or CD4$^+$ T cells ($1 \times 10^5$, $n = 5$) isolated from PBS- or SA-exposed mice were added to each well of a pre-coated 96 well plate (892199, Bio-Techne, USA) in duplicate, and stimulated them with a purified IsdB antigen (10 μg/ml) in a humidified 37 °C CO$_2$ incubator for 40 h. After incubation, the cells were removed and the plate was gently washed and incubated with 100 μl/well of a detection antibody concentrate (892200, Bio-Techne, Minneapolis, USA) for 2 h at room temperature. Then the plate was washed and incubated with 100 μl/well of Streptavidin-AP (895358, Bio-Techne, Minneapolis, USA) for 2 h at room temperature. Finally, the plate was treated with 100 μl/well of BCIP/NBT Chromogen (EL417, Bio-Techne, Minneapolis, USA), and incubated for 1 h at room temperature under constant protection from light. Then the chromogen solution was discarded and the microplate was rinsed with distilled water. The number of spots were counted in a dissection microscope and the results were expressed as spot forming cells (SFC) per million cells.

## Flow cytometry

Spleens were aseptically collected from vaccinated groups and homogenized in sterile PBS (pH 7.4). Cells were centrifuged at 400 g for 5 min, followed by RBC lysis before resuspension of splenocytes in PBS. The cells ($2 \times 10^6$) were stained with Fixable Viability Dye eFluor 780 (65-0865-14, eBioscience, USA) on ice for 30 min, followed by washing with the FACS buffer (2%FBS in PBS). Then cells were incubated with FC block (TruStain FcX; 101320, Biolegend, USA) for 10 min, followed by surface staining with fluorescently labelled Abs against CD3, CD4, CD8, CD25, CD69 and B220 on ice for 30 min. The cells were washed and resuspended in FACS buffer. For intracellular cytokine staining, surface antibody staining is followed by fixation (IC Fixation Buffer; 50-112-9058, eBiosceince, USA) at room temperature for 20 min, then washed with a permeabilization buffer (421002, Biolegend, USA) and incubation with fluorescently labelled antibodies against IFN-γ, IL-17A and IL-10 in permeabilization buffer for 1 h. Cells were washed and resuspended in FACS buffer, and run on an BD FACSCanto II Flow Cytometry. The data were analyzed with FlowJo v.10 software.

Fluorescently conjugated antibodies used in this study were: PE anti-mouse/human CD45R/B220 (103208, Biolegend, USA), APC anti-mouse CD3 (100236, Biolegend, USA), Pacific Blue anti-mouse CD4 (100427, Biolegend, USA), PerCP/Cyanine5.5 anti-mouse CD4 (100434, Biolegend, USA), PerCP/Cyanine5.5 anti-mouse CD8a (100733, Biolegend, USA), PE anti-mouse CD69 (104508, Biolegend, USA), Pacific Blue anti-mouse CD25 (102022, Biolegend, USA), PE/Cyanine7 anti-mouse IL-10 (505026, Biolegend, USA), PE anti-mouse IL-17A (506903, Biolegend, USA), PerCP/Cyanine5.5 anti-mouse IFN-γ (505822, Biolegend, USA), PE/Cyanine7 Rat IgG2b (400617, Biolegend, USA), PE Rat IgG1 (400407, Biolegend, USA), and PerCP/Cyanine5.5 Rat IgG1 (400425, Biolegend, USA). Antibodies against surface antigens were used at 1:100 dilution, and antibodies against intracellular cytokines were used at 1:50 dilution (Supplementary Table 1).

## Statistical analysis and data reproducibility

GraphPad Prism version 8 was used to analyze all data (GraphPad Software, San Diego, CA, www.graphpad.com). All statistical details, including the statistical test and number of mice used per experiment, are noted in the figure legends. In vivo data are presented as mean ± SD of biological replicates and in vitro data are presented as mean ± SD of technical replicates. Two group analysis was performed using a two-tailed non-parametric Mann–Whitney unpaired Student's T test. Comparisons of multiple groups were performed using one-way ANOVA with Tuckey's post-hoc test. In the case of missing normality, non-parametric Kruskal-Wallis one-way ANOVA was used to analyze the data. Statistical significance was assigned as ****$p \leq 0.0001$; ***$p \leq 0.001$; **$< 0.01$, *$p \leq 0.05$; $p > 0.05$: ns (not significant). Outlier analysis was performed by GraphPad Prism version 8. One outlier mouse was removed from Figs.1b, 1h, 2j, 4d, and 2 mice from Fig.1f. Most in vivo and in vitro experiments were performed two or more times and noted in the figure legends.

## Reporting summary

Further information on research design is available in the Nature Portfolio Reporting Summary linked to this article.

## Data availability

All data are included in the Supplementary Information or available from the authors, as are unique reagents used in this Article. The raw numbers for charts and graphs are available in the Source Data file whenever possible. Source data are provided with this paper.

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

## Acknowledgements

We thank S. Secore (Merck) for providing the Becker WT and IsdB/HarA mutant strains. The authors acknowledge National Institute of Health funding agency (grants R01AI127406, R01AI179098, R01AI181321 and RO1AI144694).

## Author contributions

I.A.H. and G.Y.L. conceived the idea and designed the experiments. I.A.H. carried out the in vitro *and* in vivo experiments. C.M.T., J.R.C., C.G., M.L.D., X.D., and A.A. helped with the in vivo animal experiments. J.R.C. produced mntC, IsdA, FhuD2, and sdrE clones. I.A.H. produced IsdB, pertussis toxoid-IsdB, Diphtheria toxoid-IsdB clones. B.L. and W.D. performed protein purification. I.A.H. and G.Y.L. wrote the manuscript with all other authors provided significant input.

## Competing interests

I.A.H. and G.Y.L. have filed a patent application for the use of vaccine adjuvants that mediate efficient IL-17 type-immunity. All other authors declare no competing interests.
