## [Transparent Peer Review file · Nature Communications]

Pathobiont-induced suppressive immune imprints thwart T cell vaccine responses

Corresponding Author: Dr George Liu

Version 0:

Reviewer comments:

Reviewer #1

(Remarks to the Author)

There have been a number of vaccine trials in humans with use of IspB and other bacterial cell surface molecules as vaccine targets. All have failed miserably. In the Merck trial, for example, 5X persons vaccinated died compared to non-vaccinated; the trial was stopped early for this reason. The Dunny et al. and Schlievert et al. labs have shown that the reason for these failures is likely two-fold:

1. Staphylococcus aureus aggregates to cause more serious diseases. Thus, vaccination against cell surface molecules promotes IgGs that further aggregate the organisms, leading to more serious diseases.
2. There have been two major papers published in PNAS and Nature Reviews Microbiology that indicate mice are not really good models for human infectious diseases. These papers are easily ignored by researchers wishing to do mice studies.

Having said the above, the current authors, using mouse models, suggest that vaccine failures occur because of IL10 induction, and this can be overcome by IL17A-promoting adjuvants. The data presented are scientifically sound and the manuscript well-presented.

There have also been many publications in mice suggesting cell surface vaccines, including IspB and protein A variants that have been successful in mice. These have not worked in humans. Thus, what can the authors really state about the efficacy of their strategy in humans?

Reviewer #2

(Remarks to the Author)

What are the noteworthy results?

1. CD4+ T cells from SA-infected mice inhibit IspB vaccination (and may inhibit de novo generation of other T cells).
2. Imprinted T cells make a lot of IL-10 and blockade rescues vaccine efficacy.
3. CD25+ CD4+ T cells are a major source of IL-10 and their depletion restores efficacy.
4. Imprint is overcome by IL-10 or IL-10Ra deletion on CD4+ T cells.
5. STAT3 inhibition rescues imprint.
6. T cell adjuvants (CAF01 and alumTDB) promote IL-17 secretion and rescue imprint.
7. IFN γ inhibition and IL-17KO abrogates CAF01 rescue.

Will the work be of significance to the field and related fields? How does it compare to the established literature? If the work is not original, provide relevant references.

This work has the potential to be highly significant to the SA vaccine field. Identification of a role for imprinted vaccine-resistance holds the promise of new vaccine benchmarks (i.e. being designed to overcome the imprint). Additional insight into the mechanisms by which this occurs could inform new vaccines. This work builds on findings by multiple groups (including the authors). However, a role for IL-10 in immune suppression during SA infection has long been recognized and Narita et al have already reported that IL-10 blockade restores vaccine efficacy (Infect Immun 2019;87:e00494). Thus, these findings are important largely because they give more detail on the source of IL-10 and the cells on which it acts, as well as identifying its role in the T cell imprint. The use of IL-17/IFN γ promoting adjuvants to overcome the T cell imprint is also potentially very important but has been reported (Teymournejad et al, NPJ Vaccines 2023;8:3), so this work mainly adds additional mechanistic insight in this regard. The idea that the T cell imprint neutralizes "cross-reactive threats" (i.e. immune

responses against other pathogens like virus or fungi that might clear SA) is very important but not sufficiently substantiated by the data.

Does the work support the conclusions and claims, or is additional evidence needed?

In general, most of the conclusions and claims are well supported. There are several experiments that should be repeated in order to draw firm conclusions, however – these are detailed below. However, as mentioned above, the idea that the T cell imprint neutralizes “cross-reactive threats” (i.e. immune responses against other pathogens like virus or fungi that might clear SA) is not sufficiently substantiated by the data.

Are there any flaws in data analysis, interpretation, and conclusions? Do these prohibit publication or require revision?

Data analysis appears to be appropriate. However, in several instances (detailed below) the authors draw conclusions from data that do not demonstrate clear significance. This requires revision.

Is the methodology sound? Does the work meet the expected standards in your field?

Yes, the methodology is sound and meets the expected standards.

Is there enough detail provided in the methods for the work to be reproduced?

Yes.

Description of the study:

In this interesting study, Hajam and colleagues seek to build on their prior studies that identified a non-protective immune imprint imparted by *S. aureus* infection that interferes with vaccine efficacy. They previously reported an imprint on antibody responses to vaccination, so here they seek to study if infection also inhibits T cell responses to vaccination. First (Fig 1), they provide evidence that T cells mediate imprinted vaccine-resistance via several lines of investigation. SA infection of muMT mice lacking B cells results in inhibition of vaccine efficacy, transfer of T cells (CD3+ or CD4+) promotes vaccine resistance, and depletion of CD4+ T cells partly abrogates inhibition. They also use CD45.1 and CD45.2 mice to make the argument that imprinted T cells also suppress de novo T cell development to vaccination. Next (Fig 2), they show an important role for CD25+ T cell-derived IL-10 expression in driving vaccine suppression. T cell-expressed IL-10 appears to act specifically on T cells via the IL-10Ra and requires STAT3 signaling (Fig 3). They go on to show that IL-17 stimulating adjuvants such as CAF01 and alum+TDB (a component of CAF01) can overcome suppression. Interestingly, driving responses toward IL-17 (and maybe IFN γ) production can overcome suppression even in the presence of high levels of IL-10 (Fig 5). SA suppression of vaccine responses via IL-10 appeared to also be important for other experimental SA vaccines and against other bacterial isolates and SSTI (Fig 6). Finally, they tried to simulate viral or fungal infections to demonstrate that the suppressive imprint could protect against “cross-reactive threats” (Fig 7).

Overall impression:

The strengths of the study are the tractable animal models, a clear role for IL-10 blockade in restoring vaccine suppression, and well-considered alternative possibilities that were exhaustively tested. The overall conclusions that SA suppresses T cell responses to vaccination, likely due to IL-10 expression, and that this can be overcome by targeting IL-17 responses with vaccination, are well supported by the data. However, there were several instances in which data did not support the conclusions stated by the authors. Overall, the study is unnecessarily dense, there is too much data presented, and the reader is left overwhelmed. This makes it hard to critically assess the manuscript. Also, Figure 7 is unnecessary and does not add to the manuscript – the experimental plan does not address the question being asked and the data are not really interpreted or discussed in the text. It is appreciated, however, that this is a very important question that needs to be answered.

Major critiques:

1. Based on my comments on Fig 7, the title should remove the idea of cross-reactive threats as this is an overinterpretation of the data presented.
2. These data all rely on an experimental workflow that involves 3 successive IP infections over 3 weeks, followed by 3 vaccinations over the following 3 weeks. This is a very specific model that may or may not be generalizable to natural SA exposure. Moreover, with the exception of 2 experiments in Fig 6, there is no consideration of whether the T cell imprint is durable – this raises the possibility that trained immune responses could contribute to the phenotype. These limitations should be discussed.
3. In many instances, the effect that the authors assert does not seem as strong as one would anticipate, and in fact the groups appear grossly similar, albeit with different statistical differences. For example, Fig 1b, 1h, 2c, 3d, 3e, 4e – specific examples that impact interpretation are detailed below.
4. Fig 2c – there is no significant difference between the key groups here: SA/IsdB IgG1 vs. IL-10. This is an important driver of the rest of the study, so this is a major limitation.
5. Fig 2l – These results are a bit confusing. Why does it appear that unvaccinated mice that receive T cells from IsdB/HarA KO mice have much higher CFU than those that receive T cells from WT infected mice? This seems to drive the differences in protection, since there are not major differences between mice that receive T cells from WT or KO and are then vaccinated (at least for kidney CFU). Furthermore, even if the results are true, I am not sure this demonstrates antigen specificity per se, but just that infection with the IsdB/HarA KO does not suppress vaccination with IsdB.
6. Fig 6. It is surprising that the authors did not test a combination vaccine (or at least I couldn't find it) that used the protective and inhibitory antigens, both with CAF01, to see if that would overcome inhibition and restore protection. This would be an important experiment.
7. Fig 7. I don't think this adds much, if at all, to the paper. First, use of TLR7, B-glucan, and curdlan to mimic viral and fungal infections does not really mimic these infections. In fact, an alternative interpretation could be that, since these could be

thought of as adjuvants, they were ineffective in the presence of the imprint. Regardless, these results seem tacked on to broaden relevance but are not discussed much if at all in the text (lines 288-293). The authors overinterpret these findings.

Minor critiques:

1. Vaccine efficacy is entirely quantified as spleen and kidney CFU, with no measures of illness severity. The toggling back and forth between spleen and kidney CFU, with sometimes different results between the data, makes the paper confusing at points.
2. Fig 1b – although the stats work, the overall impression is that there is not much difference in these mice when comparing the naïve Alum and AlsdB and SA-alum and AlsdB groups – in each case, there appears to be better protection in the lsdB groups.
3. Fig 1d – there are differences for spleen CFU, but not kidney, which makes it difficult to interpret.
4. Fig 1f – although a trend, there were not significant differences between vaccinated mice that rec'd T cells vs. those that did not ($p=0.1$).
5. Fig 1h – The 3 groups Alum vs. AlsdB (no t cells, CD45.1, and CD45.2) all appear to have similarly improved protection with vaccination. However, there is much more heterogeneity among T cell recipients, which raises the question of if T cell transfer was similar among mice.
6. Fig 2a – it would be helpful to look at cytokines other than IL-10 – it should not come as a surprise that splenocytes from SA-exposed and vaccinated mice would make more cytokines in response to lsdB.
7. For use of the Vert-X mice, the inability to look at IL-10 expression in innate immune cells is a limitation – it does not nullify the conclusions regarding IL-10 and T cells, but it does allow that other cells may also contribute.
8. Fig 3b – controls appear to missing – there are no naïve mice in this plot, as done in Fig 3a.
9. Fig 3d – the results are difficult to interpret here because only transfer of T cells to knockout mice is shown, but not the WT mice. Thus the Alum only knockout group has much fewer CFU, making it more difficult to detect a difference vs. vaccinated mice.
10. Fig 5a – is serum protective in naïve mice? This would be helpful to know.
11. Fig 5b (and also throughout the figure) It is interesting that CAF01 alone has some protection in some places but not others – the authors should discuss the antigen-independent vs. antigen-dependent protection.
12. Fig 6c-g. Why were these challenges so much later than the others (listed as D56 and D85)? This is a good idea to see if the effects are durable but this is not discussed.
13. Fig S4 is listed as Fig S3 in the text.

Reviewer #3

(Remarks to the Author)

The current manuscript addresses a long-standing unknown in the field of *Staphylococcus aureus* research – why so many vaccine candidates have failed in clinical trials. *S. aureus* is a common human colonizer and opportunistic pathogen that has now developed resistance to all approved antibiotics. Development of an effective anti-*S. aureus* vaccine is a critical clinical need that has proved elusive despite significant effort from the research community and pharmaceutical industry. It has recently been accepted that cellular immunity is crucial for the control of *S. aureus* infection, and research efforts are now focused on the promotion of IL-17 and IFN γ -mediated immunity. *S. aureus* has developed an arsenal of immune evasion mechanisms, however, and the circumvention of these factors will also be crucial to the design of novel vaccine formulations. It is in this context that the outcomes of the current research lie.

This manuscript helps to elucidate bacterial-driven suppressive mechanisms in the host and offers hope that novel adjuvants could circumvent this hindrance to development of an effective *S. aureus* vaccine. The authors build on previous work (Tsai et al., 2022) from the group that demonstrated how prior exposure to *S. aureus* attenuated protection from the lsdB vaccine, despite promoting robust lsdB antibody titers in trial subjects (Fowler et al., 2013). Here, they identify the generation of IL-10-expressing regulatory T cells following exposure to SA, that are re-activated following subsequent stimulation with lsdB or bacterial challenge. This response creates a suppressive environment that limits the capacity of Th1 and Th17-driven phagocytic responses and prevents clearance of the bacterium. The use of newer vaccine adjuvants that strongly promote IL-17-mediated immunity was found to overcome this effect however, while cursory explorations attempt to address a possible evolutionary rationale for the mechanisms described. The overall findings of the current research are very interesting, with a sound rationale and generally supported by well-delineated mechanistic insights. A number of concerns should be addressed, however, to improve the accuracy and legitimacy of manuscript.

1. The sheer volume of work presented, and the intricacy of many of the experiments is impressive, but is written in a way that, at times, leaves the reader to feel bamboozled into accepting the authors interpretation. Instead, an effort should be made to carefully describe the rationale for, design of and outcome of each experiment. More text should be given to explaining many of the experimental designs – particularly in the figure legends throughout the manuscript. Is there a word limit here that precludes proper description of experimental design?
2. Following on from the above, it is not clear until the last two figures at what time-points the mice were challenged post-vaccination. In the methods section, this is listed as being 7, 14, 60 or 85 d.p.v. It appears, however, that mice are challenged at the later time-points only in Figure 6 – and at 56 and 85 d.p.v. (not 60 or 85) according to the text on lines 270-271.
 - a. Challenging mice 7 and 14 d.p.v. is not a true measure of a vaccine recall response. There are likely still effector memory T cells (and IL-10+ iTregs?) in circulation at this time. The only truly antigen-recall responses in the paper are those described in Figure 6 and this tempers the validity of the earlier observed results. More should be made of this limitation in the discussion of the paper.
 - b. There is no rationale provided as to why the authors moved to this longer (more representative) vaccine protocol at this

stage of the paper, or why some experiments were measured at 56 d.p.v and others at 85 d.p.v?

3. In a similar vein, bacterial burden in peripheral tissues was measured at 20 hours post-challenge throughout the manuscript. Is this really a measure of vaccine effectiveness? Would this data stand up if these mice had been left 56+ d.p.v. before challenge, when no residual vaccine-induced immune response would be present? If the point is that Treg-derived IL-10 prevents Th17-mediated bacterial clearance, then surely a later time-point post-challenge (72h?) would have been a better measure of the effect of what is an adaptive immune response, which takes time to develop. This is exemplified by the statement in lines 131-132 that “the frequencies of CD4+T, CD8+T and B cells did not change with SA infections” ... over a 20h period!

a. Also, why is no data presented on bacterial burden at the site of infection? The approach presented by the authors is more reflective of bacterial dissemination in the hours following i.p. infection, rather than a true readout of a blunted adaptive antigen-specific immune response as currently interpreted. The rapidity of the observed effects (reduced dissemination or bacterial clearance from the spleen and kidneys??) may indicate an innate immune component or the presence of residual Teff cells as a result of the short time between vaccination and challenge (as in 2 above). Although an anti-gdTCR mAb had did not inhibit vaccine efficacy in Figure 1D, neither did the anti-CD4 mAb in the kidneys – where the vast majority of data throughout this paper is observed. This limitation and the possibility of alternate innate immune mechanisms needs to be addressed further, ideally with supporting data.

4. The early data upon which the entire premise of the paper is built, is not as clear-cut as the authors describe in the text, and is even difficult to fathom in some instances - despite the statistical data supporting the authors conclusions. For example, the non-significant effect of the Alsdb vaccine in SA-exposed mice in Figure 1B looks very close to being significant if the upper outlier was removed from the dataset. Was analysis of outliers performed to ascertain the inclusion or omission of these from the data? No mention is made of this caveat in the results or discussion sections.

Similarly, in Figure 1h and Extended Figure 1e, it looks very likely from the data that the transfer of CD4 T cells alone to recipient mice reduced bacterial burden in peripheral tissues upon challenge and the fact that there is no significant reduction in bacterial burden in Alsdb immunized mice may be due to the lower bacterial CFUs in Alum-treated mice that had also received SA-exposed CD4 T cells. Again, a likely outlier is retained in Figure 1h that may be affecting the significance of this data. Why is there a limit-of-detection in this figure but not others?

The lack of a statistically significant effect of the anti-CD69 mAb in Figure 2j is similarly questionable given the single outlier bringing up the non-protected column. Moreover, does depleting CD69+ T cells not also remove recently activated protective Teff cells??

In all cases, the results are presented as conclusive and often never again discussed. A more humble discussion of the limitations of the experimental protocols and the data as presented would strengthen my faith in the research as a whole.

5. Although data is presented supporting a role for IFNg in protection of mice following SA challenge, the clear hypothesis of the authors is that IL-17-mediated immunity is the key factor in protection against infection. This is in line with a wealth of reports on cellular immune responses to SA in mice and is supported here by the fact that CAF01 – a C-type lectin stimulating adjuvant – is the most promising of the novel adjuvants tested to overcome IL-10-mediated immunosuppression. It should be made clear in the discussion, however, that correlates of immunity in humans have been less clearly defined than in mice, and IFNg appears to be the crucial factor in macrophage-mediated control of infection in humans.

6. The statement “Abrogation of lsdB vaccine protection by transfer of CD4+T cells (Fig. 1g) suggests that vaccine interference can occur independently of T or APC depletion” on lines 116-118 is not clear and should be clarified more thoroughly. Macrophage-derived IL-10 is clearly an important immune-evasion mechanism in mice (and humans) and the short time from challenge to analysis allows for an innate component to this data.

7. Nothing is made of the fact that mice heterozygous for IL-10Ra are protected by lsdB vaccination. This is a massively surprising effect to observe and requires further clarification and discussion at the very least.

8. A better explanation of the SA-CD4 T cell adoptive transfer experiments in Figure 5 and Ext Fig 8 is required. Although CD4 T cell transfer was protective, most of the CAF01 alone data indicates a protective innate immune component to this protocol. Again, is 14 d.p.v. sufficient to call this a true antigen-recall (vaccine) response?

9. Also, why were splenocytes pooled for this experiment (and technical replicates presented)??

10. Why were IL-17A/F KO mice used in Figure 5, but only a mAb against IFNg? This makes it difficult to compare the data presented.

Moreover, IL-17 KO mice are highly susceptible to SA infection (particularly at 3×10^7 CFU). It is surprising that there was not a higher bacterial burden seen in the peripheral tissues of these mice??

No discussion is made either of the decreased SA burden in mice treated with CAF01, in the absence of IL-17A or IL-17F. What is the likely mechanism of protection in this setting?

11. The gating strategies for B220 cells and CD25+ Tregs in Ext Data Figure 2 are questionable given the low frequencies of IL-10+ cells and the inclusion of cells from a gated out population to the left of these plots. This analysis should be revisited.

12. Novel approaches to the control of *S. aureus* and other infections cannot rely solely on enhancing activation of the inflammatory response. More should be made in the discussion of the crucial balance between IL-17-driven protection and pathology in the inflammatory response to infection, particularly in light of the recently described role of Treg cells in this process (Xu et al., 2023).

Version 1:

Reviewer comments:

Reviewer #1

(Remarks to the Author)

This revised manuscript addresses vaccination against *Staphylococcus aureus*. The basis for the work is that overcoming IL10 suppression against cell surface *S. aureus* antigens is necessary and can be done by IL17A and IFN gamma adjuvanticity.

The authors have addressed my comments by adding some lines to the discussion. However, they have not really addressed by comments. *S. aureus* aggregates to cause human disease. Vaccination is likely to enhance aggregation making disease worse. The authors have made some comments on limitations, but have ignored this important observation.

Reviewer #2

(Remarks to the Author)

Overall summary:

The authors did a very nice job in responding to the many suggestions offered by the reviewers. For my major concerns, the most pertinent revisions were:

1. More mice that now demonstrate a clear difference in the important groups in Fig 2c.
2. Removal of Figure 7.
3. Demonstration that serum transfer from CAF01 vaccinated mice did not restore vaccine efficacy.
4. Multiple points which had added or more nuanced discussion – specifically, limitations of the model, stronger consideration and discussion of the role of imprinted immunity mediated by CAF01, more nuanced discussion where there were only trends, discussion of spleen vs. kidney, justification of the longer term experiments, and the use of the combination vaccine.

Overall, the manuscript is much improved and easier to follow. I do have a few suggestions that would improve it even more.

Minor comments:

1. Lines 120-124, Fig 1. If the authors want to test if transferred T cells suppress de novo generation of vaccine-specific T cells, it would be informative to quantify CD45.2 vaccine-reactive T cells by pulling out T cells and stimulating with vaccine antigen.
2. Line 151 refers to Fig 2a, but should be 3a.
3. Line 214, supp Fig 5. There were decreased cytokine (IL-17) responses to HKSA in *S. aureus* exposed mice – might this indicate that there is indeed some sort of T cell defect that was then restored?
4. Line 227, Fig 6. The logic for Fig 6 panels g-h is not well explained.

Reviewer #3

(Remarks to the Author)

The authors have adequately addressed my initial concerns, including the addition of new data (or revision of previous analysis) where appropriate and tempering of certain conclusions, and I am happy to recommend acceptance of the manuscript in its current format.

Responses to Reviewer Comments

REVIEWER COMMENTS

Reviewer #1 (Remarks to the Author):

There have been a number of vaccine trials in humans with use of IsdB and other bacterial cell surface molecules as vaccine targets. All have failed miserably. In the Merck trial, for example, 5X persons vaccinated died compared to non-vaccinated; the trial was stopped early for this reason. The Dunny et al. and Schlievert et al. labs have shown that the reason for these failures is likely two-fold:

1. Staphylococcus aureus aggregates to cause more serious diseases. Thus, vaccination against cell surface molecules promotes IgGs that further aggregate the organisms, leading to more serious diseases.
2. There have been two major papers published in PNAS and Nature Reviews Microbiology that indicate mice are not really good models for human infectious diseases. These papers are easily ignored by researchers wishing to do mice studies.

Having said the above, the current authors, using mouse models, suggest that vaccine failures occur because of IL10 induction, and this can be overcome by IL17A-promoting adjuvants. The data presented are scientifically sound and the manuscript well-presented.

There have also been many publications in mice suggesting cell surface vaccines, including IsdB and protein A variants that have been successful in mice. These have not worked in humans. Thus, what can the authors really state about the efficacy of their strategy in humans?

We thank the reviewer for her/his positive comment. We agree that until there is a successful trial, the insight gained from mouse studies will be a “best estimate” of the human response to SA vaccines. Having said that, the SA preexposure model has much greater success explaining both active and passive SA immunization trials than the standard naïve mouse model. We have added **lines 439-447** to the discussion section to underline these relevant points from the reviewer.

Reviewer #2 (Remarks to the Author):

What are the noteworthy results?

1. CD4+ T cells from SA-infected mice inhibit AIsdB vaccination (and may inhibit de novo generation of other T cells).
2. Imprinted T cells make a lot of IL-10 and blockade rescues vaccine efficacy.
3. CD25+ CD4+ T cells are a major source of IL-10 and their depletion restores efficacy.

4. Imprint is overcome by IL-10 or IL-10Ra deletion on CD4+ T cells.
5. STAT3 inhibition rescues imprint.
6. T cell adjuvants (CAF01 and alumTDB) promote IL-17 secretion and rescue imprint.
7. IFN γ inhibition and IL-17KO abrogates CAF01 rescue.

Will the work be of significance to the field and related fields? How does it compare to the established literature? If the work is not original, provide relevant references.

This work has the potential to be highly significant to the SA vaccine field. Identification of a role for imprinted vaccine-resistance holds the promise of new vaccine benchmarks (i.e. being designed to overcome the imprint). Additional insight into the mechanisms by which this occurs could inform new vaccines. This work builds on findings by multiple groups (including the authors). However, a role for IL-10 in immune suppression during SA infection has long been recognized and Narita et al have already reported that IL-10 blockade restores vaccine efficacy (Infect Immun 2019;87:e00494). Thus, these findings are important largely because they give more detail on the source of IL-10 and the cells on which it acts, as well as identifying its role in the T cell imprint. The use of IL-17/IFN γ promoting adjuvants to overcome the T cell imprint is also potentially very important but has been reported (Teymournejad et al, NPJ Vaccines 2023;8:3), so this work mainly adds additional mechanistic insight in this regard. The idea that the T cell imprint neutralizes “cross-reactive threats” (i.e. immune responses against other pathogens like virus or fungi that might clear SA) is very important but not sufficiently substantiated by the data.

We thank the reviewer for her or his positive and insightful review.

Regarding the nice study by Narita and colleagues, the research is not related to SA imprinting since the superantigen vaccine that was investigated was administered to naïve mice. Vaccine efficacy waned for unclear reason, but could be restored with anti-IL10 antibodies. Although interesting, the work is different from the imprint-focused mechanism detailed in our study.

Likewise, just prior to resubmission, we noted a new study by the McLoughlin group that demonstrated improved T cell responses to staphylococcal vaccinations by anti-IL10 treatment. While the paper suggests the applicability of anti-IL10 treatment to overcome the effect of imprinting, none of the vaccination experiments were performed in the setting of prior SA exposure. We have added the study and the Narita study to our discussion (**Lines 348-356**)

Teymournejad’s elegant paper of toxin vaccines demonstrated interference by immune imprinting, although the proposed mechanism is unrelated to IL10. The study clearly demonstrated the utility of IL-17/IFN γ promoting adjuvants in overcoming T cell imprints. Notably, anti-toxin imprints are protective in our i.p. systemic infection model through antibody neutralization of toxins, whereas anti-CWA (cell-wall anchored) antigen imprints are non-protective (Caldera Cell Rep Med 2024). It is unclear if anti-toxin imprints release IL10, but IL10 clearly plays a central role with suppression of CWA vaccines through inhibition of Th17 development. Please see added discussion **on lines 379-388**.

We agree that the cross-reactive threats are insufficiently substantiated, and have therefore removed figure 7 and claim of having demonstrated this mechanism until we could provide more robust corroboration in a future manuscript, as this will require substantial additional work.

Does the work support the conclusions and claims, or is additional evidence needed?

In general, most of the conclusions and claims are well supported. There are several experiments that should be repeated in order to draw firm conclusions, however – these are detailed below. However, as mentioned above, the idea that the T cell imprint neutralizes “cross-reactive threats” (i.e. immune responses against other pathogens like virus or fungi that might clear SA) is not sufficiently substantiated by the data.

We agree that we have insufficient data to robustly support a cross-reactive threat hypothesis and have removed figure 7.

Are there any flaws in data analysis, interpretation, and conclusions? Do these prohibit publication or require revision?

Data analysis appears to be appropriate. However, in several instances (detailed below) the authors draw conclusions from data that do not demonstrate clear significance. This requires revision.

Please see specific revisions in the sections below.

Is the methodology sound? Does the work meet the expected standards in your field?

Yes, the methodology is sound and meets the expected standards.

Is there enough detail provided in the methods for the work to be reproduced?

Yes.

Description of the study:

In this interesting study, Hajam and colleagues seek to build on their prior studies that identified a non-protective immune imprint imparted by *S. aureus* infection that interferes with vaccine efficacy. They previously reported an imprint on antibody responses to vaccination, so here they seek to study if infection also inhibits T cell responses to vaccination. First (Fig 1), they provide evidence that T cells mediate imprinted vaccine-resistance via several lines of investigation. SA infection of muMT mice lacking B cells results in inhibition of vaccine efficacy, transfer of T cells (CD3+ or CD4+) promotes vaccine resistance, and depletion of CD4+ T cells partly abrogates inhibition. They also use CD45.1 and CD45.2 mice to make the argument that imprinted T cells also suppress de novo T cell development to vaccination. Next (Fig 2), they show an important role for CD25+ T cell-derived IL-10 expression in driving vaccine suppression. T cell-expressed IL-10 appears to act specifically on T cells via the IL-10Ra and requires STAT3 signaling (Fig 3). They go on to show that IL-17 stimulating adjuvants such as CAF01 and alum+TDB (a component of CAF01) can overcome suppression. Interestingly, driving responses toward IL-17 (and maybe IFNg) production can overcome suppression even in the presence of high levels of IL-10 (Fig 5). SA suppression of vaccine responses via IL-10 appeared to also be important for other experimental SA vaccines and against other bacterial isolates and SSTI (Fig 6). Finally, they tried to simulate viral or fungal infections to demonstrate that the suppressive imprint could protect against “cross-

reactive threats” (Fig 7).

Overall impression:

The strengths of the study are the tractable animal models, a clear role for IL-10 blockade in restoring vaccine suppression, and well-considered alternative possibilities that were exhaustively tested. The overall conclusions that SA suppresses T cell responses to vaccination, likely due to IL-10 expression, and that this can be overcome by targeting IL-17 responses with vaccination, are well supported by the data. However, there were several instances in which data did not support the conclusions stated by the authors. Overall, the study is unnecessarily dense, there is too much data presented, and the reader is left overwhelmed. This makes it hard to critically assess the manuscript. Also, Figure 7 is unnecessary and does not add to the manuscript – the experimental plan does not address the question being asked and the data are not really interpreted or discussed in the text. It is appreciated, however, that this is a very important question that needs to be answered.

We have carefully revised our manuscript to address instances where the data did not support the conclusions (detailed below). In addition, we have significantly modified the figures, figure legends and text throughout to make the manuscript easier to read (please see markings throughout the text and figure legends. Few of the figures that are difficult to interpret and that are non-contributory to the main conclusions are removed (please see below). Figure 7 fits into this description and has been removed.

Major critiques:

1. Based on my comments on Fig 7, the title should remove the idea of cross-reactive threats as this is an overinterpretation of the data presented.

We have removed the idea of cross-reactive threats from the title and abstract.

2. These data all rely on an experimental workflow that involves 3 successive IP infections over 3 weeks, followed by 3 vaccinations over the following 3 weeks. This is a very specific model that may or may not be generalizable to natural SA exposure.

We have published several models of SA-exposure (one IV SA infection followed by vancomycin treatment, 2-3 i.p. infection, and soft tissue infections). We showed that in each case, prior SA exposure leads to IsdB vaccine interference (Tsai 2022). Key to the induction of interference appears to be having sufficient imprint and the type of imprint (Caldera 2024). However, prior SA exposure is likely quite varied in humans. We thus acknowledged the potential limitation of our model that relies on 3 IP weekly infections (**lines 441-445**).

Moreover, with the exception of 2 experiments in Fig 6, there is no consideration of whether the T cell imprint is durable – this raises the possibility that trained immune responses could contribute to the phenotype. These limitations should be discussed.

Innate training was an important consideration and prompted us to rely on adoptive transfer of CD3+ or CD4+T cells to corroborate the role of T cells.

In the case of CAF01 adjuvancy, we agree that innate training likely contributed to protection in shorter term experiments. This finding led us to perform additional SA challenge beyond 50 dpv to delineate separation of innate and antigen-dependent protection. We have added this rationale on lines **227-230, 253-255, 449-452**.

3. In many instances, the effect that the authors assert does not seem as strong as one would anticipate, and in fact the groups appear grossly similar, albeit with different statistical differences. For example, Fig 1b, 1h, 2c, 3d, 3e, 4e – specific examples that impact interpretation are detailed below.

We have revisited each of the examples above. For those instances where the “eye test” does not support the statistics, we softened the conclusion drawn from the experiments (please see below). We have also added more data to make fig 2c more robust and removed fig. 3d since the experiment is difficult to interpret and is non-essential to the main conclusions of the paper.

4. Fig 2c – there is no significant difference between the key groups here: SA/AlsdB IgG1 vs. IL-10. This is an important driver of the rest of the study, so this is a major limitation.

We increased the sample size of the experiments. With the new added data, the difference between SA/AlsdB IgG1 vs IL10 is now significant (**Fig. 2c**).

5. Fig 2l – These results are a bit confusing. Why does it appear that unvaccinated mice that receive T cells from lsdB/HarA KO mice have much higher CFU than those that receive T cells from WT infected mice? This seems to drive the differences in protection, since there are not major differences between mice that receive T cells from WT or KO and are then vaccinated (at least for kidney CFU). Furthermore, even if the results are true, I am not sure this demonstrates antigen specificity per se, but just that infection with the lsdB/HarA KO does not suppress vaccination with lsdB.

We performed statistical comparison of unvaccinated mice that receive T cells from lsdB/HarA KO to those that received T cells from WT mice. Neither one (from spleen or kidneys) achieved significance (>0.8 for the kidneys and >0.13 for the spleen). The data appear more robust in the spleen than kidneys. But we agree with the reviewer that the experimental setup does not provide proof of antigen specificity. Hence, we softened our conclusion and stated that our findings suggest antigen specificity (**Lines 177-181**).

6. Fig 6. It is surprising that the authors did not test a combination vaccine (or at least I couldn't find it) that used the protective and inhibitory antigens, both with CAF01, to see if that would overcome inhibition and restore protection. This would be an important experiment.

We have performed the CAF01 experiment as suggested. Although CAF01 conferred protection on its own as per prior experiments, CAF01 significantly (albeit moderately) improved the efficacy of the combination vaccine above the level of CAF01, unlike alum (**Fig. 6c, Sup Fig 10b**)

7. Fig 7. I don't think this adds much, if at all, to the paper. First, use of TLR7, B-glucan, and curdlan to mimic viral and fungal infections does not really mimic these infections. In fact, an alternative interpretation could be that, since these could be thought of as adjuvants, they were ineffective in the presence of the imprint. Regardless, these results seem tacked on to broaden relevance but are not discussed much if at all in the text (lines 288-293). The authors overinterpret these findings.

We agree and have accordingly removed figure 7.

Minor critiques:

1. Vaccine efficacy is entirely quantified as spleen and kidney CFU, with no measures of illness severity. The toggling back and forth between spleen and kidney CFU, with sometimes different results between the data, makes the paper confusing at points.

Alum adjuvanted vaccine efficacy was evaluated by LD90 in our original manuscript (Tsai 2022), and we agree that relying on CFU burden is a limitation of the study (**noted now on lines 447-448**).

We think that the CFU difference being significant in one organ and not the other reflects the smaller magnitude of T cell responses compared to whole T and B cell responses to lsdB vaccine of our prior papers. For the most important phenotypes, the different approaches we have taken was intended to make sure that our conclusions are largely robust in spite of the smaller magnitude of T cell findings. In new discussions, we have remarked on these limitations (**line 455-459**).

2. Fig 1b – although the stats work, the overall impression is that there is not much difference in these mice when comparing the naïve Alum and lsdB and SA-alum and lsdB groups – in each case, there appears to be better protection in the lsdB groups.

Thanks. We reviewed the data from figure 1b and removed the outlier based on statistics (graphpad) as suggested by reviewer 3. Even then, approximately 75% of data points from the two groups were roughly overlapping. The significance was at 0.0049 for Alum versus lsdB control, and 0.2894 for SA-Alum vs SA-lsdB. Thus, we think that this set of data does demonstrate lack of vaccine efficacy in SA-exposed MuMt- mice, compared to the controls. This is supported by similar conclusions drawn from the adoptive CD3+ transfer experiment (Figure 1c). In addition, we have included data from another (TLR7 adjuvanted) lsdB vaccine also performed in MuMt- mice (Figure 1d) that also demonstrates interference with T cell vaccines by SA imprints.

3. Fig 1d – there are differences for spleen CFU, but not kidney, which makes it difficult to interpret.

We agree. We cannot clearly define the population within CD3+T cells that is responsible for protection in the kidneys (now in Supp. Fig1g). This is now reflected in our more nuanced discussion on lines **96-99**.

4. Fig 1f – although a trend, there were not significant differences between vaccinated mice that rec'd T cells vs. those that did not ($p=0.1$).

We performed “outlier” analysis (graphpad). Upon removal of the outlier in the Alum-IsdB group, significance (<0.05) is demonstrated between the vaccinated mice that received or did not receive the CD4+ T cells (Fig. 1f).

5. Fig 1h – The 3 groups Alum vs. AlsdB (no t cells, CD45.1, and CD45.2) all appear to have similarly improved protection with vaccination. However, there is much more heterogeneity among T cell recipients, which raises the question of if T cell transfer was similar among mice.

The T cell transfers were done the same way, but we agree that the data appear not as clearcut as suggested by the statistics. Accordingly, we have provided a more nuanced discussion and conclusion (lines 125-126).

6. Fig 2a – it would be helpful to look at cytokines other than IL-10 – it should not come as a surprise that splenocytes from SA-exposed and vaccinated mice would make more cytokines in response to IsdB.

Cytokine profiling is looked at in a separate manuscript that is in submission. Notably, however, we studied IL-6 induced by SA in this manuscript because of IL6's link to IL10 and STAT3. Blocking of IL-6 had no effect on vaccine efficacy (Figure 3d).

7. For use of the Vert-X mice, the inability to look at IL-10 expression in innate immune cells is a limitation – it does not nullify the conclusions regarding IL-10 and T cells, but it does allow that other cells may also contribute.

IL10 production by various innate cells with SA infection is well established and discussed in our manuscript. Related to the role of IL10 secreted by myeloid cells in vaccine suppression, we showed using lineage specific IL10 KO that myeloid (LysM) derived IL10 is less important than the CD4 source of IL10 for the modulation of vaccine responses in SA-exposed mice (Figure 3a and b).

8. Fig 3b – controls appear to missing – there are no naïve mice in this plot, as done in Fig 3a.

We have shown that IL10 production with the control Alum and Alum IsdB are modest, and increases substantially with SA reinfection plus vaccination (Fig. 2a). Hence, we did not evaluate IsdB vaccination of naïve CD4-IL10^{-/-} mice, although this would have been an informative control.

9. Fig 3d – the results are difficult to interpret here because only transfer of T cells to knockout mice is shown, but not the WT mice. Thus, the Alum only knockout group has much fewer CFU, making it more difficult to detect a difference vs. vaccinated mice.

Thanks for pointing this out. We do have a group of SA-CD4⁺T cell transfer control that was not previously added (**please see above figure**). However in rethinking this experiment, we realized that the anticipated results could be more complex than previously perceived: While the transfer of SA-CD4⁺T cells introduce abundant IL10 that would have no effect on CD4-IL10R KO, the transferred non-protective T cells could nonetheless compete for antigen presentation and make the results more difficult to interpret. As the experiment is not essential for the manuscript, and difficult to interpret, we opted to remove the experiment.

10. Fig 5a – is serum protective in naïve mice? This would be helpful to know.

Serum from naïve mice immunized with Alum/IsdB is protective (Tsai 2022). Serum from naïve CAF01 IsdB immunized mice is modestly protective in kidneys but not in the spleen (**Suppl Fig. 8a**)

11. Fig 5b (and also throughout the figure) It is interesting that CAF01 alone has some protection in some places but not others – the authors should discuss the antigen-independent vs. antigen-dependent protection.

We agree that CAF01 appears to induce antigen-independent protection likely from innate training, in shorter term experiments. This is now discussed on **lines 227-229** and **251-255**.

12. Fig 6c-g. Why were these challenges so much later than the others (listed as D56 and D85)? This is a good idea to see if the effects are durable but this is not discussed.

These challenges are intentionally delayed because of likely innate training that we noted with CAF01 alone at earlier time points. We have added a rationale for evaluating these challenges at

the later time point - to evaluate the durability of protection and separate antigen-independent from antigen-dependent responses (please see line 300-303).

13. Fig S4 is listed as Fig S3 in the text.

Corrected

Reviewer #3 (Remarks to the Author):

The current manuscript addresses a long-standing unknown in the field of *Staphylococcus aureus* research – why so many vaccines candidates have failed in clinical trials. *S. aureus* is a common human colonizer and opportunistic pathogen that has now developed resistance to all approved antibiotics. Development of an effective anti-*S. aureus* vaccine is a critical clinical need that has proved elusive despite significant effort from the research community and pharmaceutical industry. It has recently been accepted that cellular immunity is crucial for the control of *S. aureus* infection, and research efforts are now focused on the promotion of IL-17 and IFN γ -mediated immunity. *S. aureus* has developed an arsenal of immune evasion mechanisms, however, and the circumvention of these factors will also be crucial to the design of novel vaccine formulations. It is in this context that the outcomes of the current research lie.

This manuscript helps to elucidate bacterial-driven suppressive mechanisms in the host and offers hope that novel adjuvants could circumvent this hindrance to development of an effective *S. aureus* vaccine. The authors build on previous work (Tsai et al., 2022) from the group that demonstrated how prior exposure to *S. aureus* attenuated protection from the IsdB vaccine, despite promoting robust IsdB antibody titers in trial subjects (Fowler et al., 2013). Here, they identify the generation of IL-10-expressing regulatory T cells following exposure to SA, that are re-activated following subsequent stimulation with IsdB or bacterial challenge. This response creates a suppressive environment that limits the capacity of Th1 and Th17-driven phagocytic responses and prevents clearance of the bacterium. The use of newer vaccine adjuvants that strongly promote IL-17-mediated immunity was found to overcome this effect however, while cursory explorations attempt to address a possible evolutionary rationale for the mechanisms described. The overall findings of the current research are very interesting, with a sound rationale and generally supported by well-delineated mechanistic insights. A number of concerns should be addressed, however, to improve the accuracy and legitimacy of manuscript.

We appreciate the reviewer's constructive comments and suggestions for improving our manuscript.

1. The sheer volume of work presented, and the intricacy of many of the experiments is impressive, but is written in a way that, at times, leaves the reader to feel bamboozled into accepting the authors interpretation. Instead, an effort should be made to carefully describe the rationale for, design of and outcome of each experiment. More text should be given to explaining many of the

experimental designs – particularly in the figure legends throughout the manuscript. Is there a word limit here that precludes proper description of experimental design?

Thank you. We have carefully revised our manuscript, added rationale, design, and outcome throughout the text. Significantly, we have added captions within the figures to help orient the reader, while adding details of the experiments in the figure legends as the word limit permits.

2. Following on from the above, it is not clear until the last two figures at what time-points the mice were challenged post-vaccination. In the methods section, this is listed as being 7, 14, 60 or 85 d.p.v. It appears, however, that mice are challenged at the later time-points only in Figure 6 – and at 56 and 85 d.p.v. (not 60 or 85) according to the text on lines 270-271.

The post-challenge is described in some of the schematics. We have gone back and more extensively annotated the timing of post vaccine challenges in the figure legends and in the method section (lines 554-555).

a. Challenging mice 7 and 14 d.p.v. is not a true measure of a vaccine recall response. There are likely still effector memory T cells (and IL-10+ iTregs?) in circulation at this time. The only truly antigen-recall responses in the paper are those described in Figure 6 and this tempers the validity of the earlier observed results. More should be made of this limitation in the discussion of the paper.

Thanks for the suggestion. We have noted this limitation in our discussion (lines 449-452).

b. There is no rationale provided as to why the authors moved to this longer (more representative) vaccine protocol at this stage of the paper, or why some experiments were measured at 56 d.p.v and others at 85 d.p.v?

We noted in early CAF01 studies that CAF01 has antigen-independent protective effect (most likely innate training). To assess durability and separate antigen-independent and antigen-dependent effect, we proceeded with additional evaluation on day > 50 post vaccination. The rationale has now been added to lines 300-303. There was no clear rationale for the choice of D56 versus D85. We agree that selecting a single date would have been better.

3. In a similar vein, bacterial burden in peripheral tissues was measured at 20 hours post-challenge throughout the manuscript. Is this really a measure of vaccine effectiveness? Would this data stand up if these mice had been left 56+ d.p.v. before challenge, when no residual vaccine-induced immune response would be present? If the point is that Treg-derived IL-10 prevents Th17-mediated bacterial clearance, then surely a later time-point post-challenge (72h?) would have been a better measure of the effect of what is an adaptive immune response, which takes time to develop. This is exemplified by the statement in lines 131-132 that “the frequencies of CD4+T, CD8+T and B cells did not change with SA infections” ... over a 20h period!

The point is well taken and we agree that this is a limitation of the study. Although we could not repeat all experiments, we did evaluate vaccine response time point to 48h, and there is indeed increased difference at 48 h over 24h (**Supplementary Fig. 1d-e**). These limitations are discussed on **lines 88-91 and 448-449**.

a. Also, why is no data presented on bacterial burden at the site of infection? The approach presented by the authors is more reflective of bacterial dissemination in the hours following i.p. infection, rather than a true readout of a blunted adaptive antigen-specific immune response as currently interpreted. The rapidity of the observed effects (reduced dissemination or bacterial clearance from the spleen and kidneys??) may indicate an innate immune component or the presence of residual Treg cells as a result of the short time between vaccination and challenge (as in 2 above).

Following the reviewer's suggestion, we have enumerated the bacterial burden at the site of infection at 48h (peritoneum, **Supplementary Fig. 1f**). Clear protection was shown.

We agree with the critique regarding the possible role of innate immune component, although the focus is less on excluding the role of innate immune components. Please see **lines 452-455** for added discussion.

Although an anti-gdTCR mAb had did not inhibit vaccine efficacy in Figure 1D, neither did the anti-CD4 mAb in the kidneys – where the vast majority of data throughout this paper is observed. This limitation and the possibility of alternate innate immune mechanisms needs to be addressed further, ideally with supporting data.

Thank you. In our revised manuscript we have more carefully delineated and separated the characterization of protective T cells, and suppressor T cells that interfered with vaccination.

The data pointed to the protective role of CD3+T cells with IsdB vaccination, although we were not able to corroborate the role of CD4+T cells, at least not in the kidneys. Within the kidneys, there could be multiple T cell populations, none of which alone are sufficient to protect. Hence, we agree that the role of innate immune gamma delta T cell immunity could not be excluded. This is accordingly discussed (**lines 96-99**).

Our investigation of the suppressive role of IL10 producing CD4+T cells is more comprehensive and included the use of myeloid lineage restricted IL10 KO (Fig. 3a).

4. The early data upon which the entire premise of the paper is built, is not as clear-cut as the authors describe in the text, and is even difficult to fathom in some instances - despite the statistical data supporting the authors conclusions. For example, the non-significant effect of the Alsdb vaccine in SA-exposed mice in Figure 1B looks very close to being significant if the upper outlier was removed from the dataset. Was analysis of outliers performed to ascertain the inclusion or omission of these from the data? No mention is made of this caveat in the results or discussion sections.

Thanks. We reviewed the data from figure 2b, performed and removed the outlier based on statistics. Even then, there was 75% of data points from the two groups that were overlapping. The significance was at 0.0049 for Alum versus AlsdB and 0.2894 for SA-Alum vs SA-AlsdB. Thus, we think that this set of data (unlike some of later data) does demonstrate lack of vaccine efficacy in SA-exposed MuMt- mice. This is corroborated by similar conclusions drawn from the adoptive CD3+ transfer experiment (Figure 1c).

Similarly, in Figure 1h and Extended Figure 1e, it looks very likely from the data that the transfer of CD4 T cells alone to recipient mice reduced bacterial burden in peripheral tissues upon challenge and the fact that there is no significant reduction in bacterial burden in AlsdB immunized mice may be due to the lower bacterial CFUs in Alum-treated mice that had also received SA-exposed CD4 T cells. Again, a likely outlier is retained in Figure 1h that may be affecting the significance of this data. Why is there a limit-of-detection in this figure but not others?

We have removed one outlier based on additional analysis (graphpad). Although significance is not changed, we agree that, visually, the data are not as clear cut as suggested simply by the statistics. This could reflect a modest phenotype. We noted this point in our new discussion (**lines 125-126**). The limit-of-detection is removed.

The lack of a statistically significant effect of the anti-CD69 mAb in Figure 2j is similarly questionable given the single outlier bringing up the non-protected column. Moreover, does depleting CD69+ T cells not also remove recently activated protective Teff cells??

Although the single data point did not meet the statistical definition of “outlier”, we agree with the reviewer that, by the “eye test”, it appears that CD69+T cells could have a role. The consideration of CD69 expressing Teff cells is important. Accordingly, we have provided a nuanced discussion of the role of CD69+T reg cells (**lines 170-173**).

In all cases, the results are presented as conclusive and often never again discussed. A more humble discussion of the limitations of the experimental protocols and the data as presented would strengthen my faith in the research as a whole.

We agree with the approach. We have thus carefully reevaluated these data and provided more nuanced conclusions.

5. Although data is presented supporting a role for IFN γ in protection of mice following SA challenge, the clear hypothesis of the authors is that IL-17-mediated immunity is the key factor in

protection against infection. This is in line with a wealth of reports on cellular immune responses to SA in mice and is supported here by the fact that CAF01 – a C-type lectin stimulating adjuvant – is the most promising of the novel adjuvants tested to overcome IL-10-mediated immunosuppression. It should be made clear in the discussion, however, that correlates of immunity in humans have been less clearly defined than in mice, and IFN γ appears to be the crucial factor in macrophage-mediated control of infection in humans.

Thank you for the excellent point. This has now been added to **lines 388-391**.

6. The statement “Abrogation of IsdB vaccine protection by transfer of CD4⁺T cells (Fig. 1g) suggests that vaccine interference can occur independently of T or APC depletion” on lines 116-118 is not clear and should be clarified more thoroughly. Macrophage-derived IL-10 is clearly an important immune-evasion mechanism in mice (and humans) and the short time from challenge to analysis allows for an innate component to this data.

The statement is an inference drawn from the sentence above: Various staphylococcal T modulatory mechanisms have been described, including toxin-mediated killing of CD4⁺T and antigen-presenting cells (APCs). We intended to denote that interference was likely unrelated to “toxin” killing of T cells or APC. More likely, it involves a suppressive mechanism such as that demonstrated through the transfer of IL10 expressing CD4⁺T cells (**see modified lines 135-138**).

Although we suspected that macrophage derived IL10 could play a role, vaccination of SA-exposed MIL10^{-/-} mice was not protective. While this does not preclude a suppressive role for myeloid cell derived IL10, it implies that IL10 from the myeloid cells may not be as important as IL10 from CD4⁺T cells (Fig. 3a and b).

7. Nothing is made of the fact that mice heterozygous for IL-10Ra are protected by IsdB vaccination. This is a massively surprising effect to observe and requires further clarification and discussion at the very least.

We agree that the data was surprising. An explanation would be that the amount of signal delivered through ligation of IL10 to het IL10R is somehow insufficient to inhibit protective T cell function. This would be difficult to corroborate without significant additional experimentations. Given that the finding has little bearing on conclusions drawn from the study and is likely to generate more speculation, we opted to remove the heterozygous IL10Ra data.

8. A better explanation of the SA-CD4 T cell adoptive transfer experiments in Figure 5 and Ext Fig 8 is required. Although CD4 T cell transfer was protective, most of the CAF01 alone data indicates a protective innate immune component to this protocol. Again, is 14 d.p.v. sufficient to call this a true antigen-recall (vaccine) response?

We agree that the CAF01 alone data across several experiments point to an antigen-independent effect, most likely reflecting innate immunity. This is now noted more clearly on **lines 227-229, 251-255**. However, adoptive transfer of SA-CD4 T cells was only protective with the CAF01IsdB immunized mice and not with the adjuvant alone, indicating that CAF01 alone did not induce

protection through CD4+T cells. This separation of antigen-independent (likely innate training) from antigen-dependent protection becomes more clear-cut after 50 dpv. We have also added a discussion of this issue, including the caution regarding calling this an antigen-recall response (**lines 449-452**).

9. Also, why were splenocytes pooled for this experiment (and technical replicates presented)?

We appreciate the reviewer's comments regarding pooling. For rigor, we performed the experiments independently 2-3 times.

10. Why were IL-17A/F KO mice used in Figure 5, but only a mAb against IFN γ ? This makes it difficult to compare the data presented.

We initially performed the IL17A KO experiment based on the finding that IL17 producing adjuvants confer protection in SA-exposed mice. We agree that subsequent testing of IFN γ blocking experiments should have been done alongside anti-IL17A experiments, which are included here. Here, we note that the anti-IL17A antibody effect is less striking than that noted in KO mice, with only a trend but no significance noted in the kidneys (unclear why), compared to the KO mice where there was a difference both in the kidneys and spleen. Based on the new data (Supplementary Fig. 9d), we conclude that, although CAF01 promote IsdB vaccine protection via Ila7A, the role of IFN γ is clearly important, if not more important than IL17A for CAF01 (**lines 273-276**).

Moreover, IL-17 KO mice are highly susceptible to SA infection (particularly at 3×10^7 CFU). It is surprising that there was not a higher bacterial burden seen in the peripheral tissues of these mice??

We don't know, but speculate the following: With SA infection alone, we would expect CFU to be higher in IL17KO as we and others reported previously using these mice (Sanchez Cell Host Microbe 2017). However, all groups tested here received either Alum or CAF01, all of which induce some level of IL10. IL10 antagonizes IL17 and may play a role for why we don't see as clear a difference in CFUs between WT and IL17KO mice as published. But it's a speculation.

No discussion is made either of the decreased SA burden in mice treated with CAF01, in the absence of IL-17A or IL-17F. What is the likely mechanism of protection in this setting?

CAF01 associated CFU in WT and IL17A KO mice were not different ($p=0.7638$ for kidneys, and $p>0.9$ for spleen), whereas CAF01IsdB associated CFU in WT and IL17KO mice was different ($p=0.0002$ for spleen and $p=0.05$ for kidneys). This is consistent with the innate training mechanism proposed with CAF01 (**lines 268-271**).

11. The gating strategies for B220 cells and CD25+ Tregs in Ext Data Figure 2 are questionable given the low frequencies of IL-10+ cells and the inclusion of cells from a gated out population to the left of these plots. This analysis should be revisited.

Gating strategy has been revised. Please see **Supplementary Fig. 2c**.

12. Novel approaches to the control of *S. aureus* and other infections cannot rely solely on enhancing activation of the inflammatory response. More should be made in the discussion of the crucial balance between IL-17-driven protection and pathology in the inflammatory response to infection, particularly in light of the recently described role of Treg cells in this process (Xu et al., 2023).

Please see expanded discussion of this topic on **lines 362-371**.

REVIEWER COMMENTS

Reviewer #1 (Remarks to the Author):

This revised manuscript addresses vaccination against *Staphylococcus aureus*. The basis for the work is that overcoming IL10 suppression against cell surface *S. aureus* antigens is necessary and can be done by IL17A and IFN gamma adjuvanticity.

The authors have addressed my comments by adding some lines to the discussion. However, they have not really addressed by comments. *S. aureus* aggregates to cause human disease. Vaccination is likely to enhance aggregation making disease worse. The authors have made some comments on limitations, but have ignored this important observation.

We apologize for this oversight and have now added the following sentences to the text: It's been shown that pathogen aggregation by IgGs can lead to more severe endocarditis. Thus, vaccination against SA cell surface molecules has the potential to promote microbial aggregation and contribute to more severe infection (71) (**Lines 448-450**).

Reviewer #2 (Remarks to the Author):

Overall summary:

The authors did a very nice job in responding to the many suggestions offered by the reviewers. For my major concerns, the most pertinent revisions were:

1. More mice that now demonstrate a clear difference in the important groups in Fig 2c.
2. Removal of Figure 7.
3. Demonstration that serum transfer from CAF01 vaccinated mice did not restore vaccine efficacy.
4. Multiple points which had added or more nuanced discussion – specifically, limitations of the model, stronger consideration and discussion of the role of imprinted immunity mediated by CAF01, more nuanced discussion where there were only trends, discussion of spleen vs. kidney, justification of the longer term experiments, and the use of the combination vaccine.

Overall, the manuscript is much improved and easier to follow. I do have a few suggestions that would improve it even more.

Minor comments:

1. Lines 120-124, Fig 1. If the authors want to test if transferred T cells suppress de novo generation of vaccine-specific T cells, it would be informative to quantify CD45.2 vaccine-reactive T cells by pulling out T cells and stimulating with vaccine antigen.

This is a great idea that would be supportive of suppression. This was a question we contemplated as we aimed to determine if adjuvant reversal of interference is achieved through reprogramming of non-protective imprints (plasticity) versus recruitment of de novo protective T cells. However, we do not have frozen samples that allow us to do the experiment readily. A repeat of the experiment will delay publication by 3-4 months. Hence, our preference would be to do that in a future study along with the mechanistic extension of our adjuvant studies.

2. Line 151 refers to Fig 2a, but should be 3a.

Corrected

3. Line 214, supp Fig 5. There were decreased cytokine (IL-17) responses to HKSA in *S. aureus* exposed mice – might this indicate that there is indeed some sort of T cell defect that was then restored?

Yes, that would be a more correct interpretation of the data. Accordingly, we have modified the sentence as follows: CD4⁺T cells from SA-exposed, IsdB-vaccinated mice produced higher IL-17A responses to IsdB but reduced IL-17A to HKSA, when compared to CD4⁺T cells from SA-exposed control mice. Anti-IL-10 treatment of the vaccinated mice significantly augmented CD4⁺T cell secretion of IL-17A with either antigenic stimulation, but had little effect on IFN- γ responses (**Line 213-217**).

4. Line 227, Fig 6. The logic for Fig 6 panels g-h is not well explained.

We have added the following lines to explain the results of Fig 6g-h - only adjuvants that induced the highest levels of IL-17A (CAF01 and ATDB), in combination with IsdB, protected SA-exposed mice (Fig. 4 c, e, and Supplementary Fig. 6 b and e), whereas adjuvants that induced lower levels of IL-17A (TDB, b-glucan and TLR7) were not protective (Supplementary Fig. 6 c, g, and h) (**lines 225-228**).

Reviewer #3 (Remarks to the Author):

The authors have adequately addressed my initial concerns, including the addition of new data (or revision of previous analysis) where appropriate and tempering of certain conclusions, and I am happy to recommend acceptance of the manuscript in its current format.